# Tensor Wheel Decomposition and Its Tensor Completion Application

**Zhong-Cheng Wu**[1]**, Ting-Zhu Huang**[*,1]**, Liang-Jian Deng**[*,1]**, Hong-Xia Dou**[2]**, Deyu Meng**[3,4]

[1]School of Mathematical Sciences, University of Electronic Science and Technology of China
[2]School of Science, Xihua University
[3]School of Mathematics and Statistics, Xi'an Jiaotong University
[4]Pazhou Laboratory (Huangpu)
`wuzhch97@163.com, tingzhuhuang@126.com, liangjian.deng@uestc.edu.cn`
`hongxia.dou@mail.xhu.edu.cn, dymeng@mail.xjtu.edu.cn`

## Abstract

Recently, tensor network (TN) decompositions have gained prominence in computer vision and contributed promising results to high-order data recovery tasks. However, current TN models are rather being developed towards more intricate structures to pursue incremental improvements, which instead leads to a dramatic increase in rank numbers, thus encountering laborious hyper-parameter selection, especially for higher-order cases. In this paper, we propose a novel TN decomposition, dubbed tensor wheel (TW) decomposition, in which a high-order tensor is represented by a set of latent factors mapped into a specific wheel topology. Such decomposition is constructed starting from analyzing the graph structure, aiming to more accurately characterize the complex interactions inside objectives while maintaining a lower hyper-parameter scale, theoretically alleviating the above deficiencies. Furthermore, to investigate the potentiality of TW decomposition, we provide its one numerical application, i.e., tensor completion (TC), yet develop an efficient proximal alternating minimization-based solving algorithm with guaranteed convergence. Experimental results elaborate that the proposed method is significantly superior to other tensor decomposition-based state-of-the-art methods on synthetic and real-world data, implying the merits of TW decomposition. The code is available at: *https://github.com/zhongchengwu/code_TWDec*.

## 1   Introduction

Tensors, as a higher-order generalization of matrices, can naturally preserve the complex interactions inside high-order tensor data, e.g., multispectral images (MSIs), color videos (CVs), and hyperspectral videos (HSVs). Tensor decompositions refer to the approximation of high-order tensor data by using latent factors, which can be considered as the principal components or features, leading to the reduced storage cost and computational complexity in large-scale and intractable tensor data analysis. Previously, different tensor decompositions have been proposed and widely applied in various fields, e.g., signal processing [6, 7, 19], machine learning [25, 29, 1], and medical imaging [31]. Among them, tensor singular value decomposition (t-SVD) [16], especially CANDECOMP/PARAFAC (CP) [13] and Tucker [26] decompositions, are the most classical and widely investigated [17, 34].

More recently, tensor network (TN) decompositions [27, 9, 14, 11] have consecutively been involved in computational mathematics and numerical analysis, and exhibit superior properties, such as super compressing and efficient computing [24, 15]. Particularly, some of TN models have been introduced

---

[*]Corresponding authors.

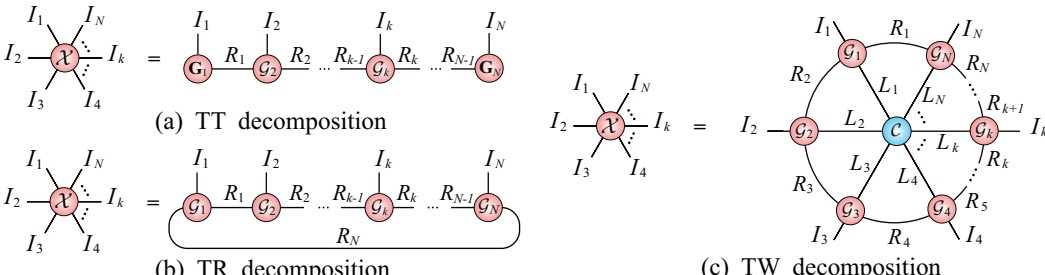

Figure 1: Graphical illustration of TT decomposition, TR one, and the proposed TW one.

to the computer vision community as powerful tools for high-order data recovery applications owing to their excellent characterization capabilities. The popular tensor train (TT) decomposition [22] is one such decomposition, in which an $N$th-order tensor is expressed as strictly ordered multi-linear products over one matrix, followed by $N$-2 third-order tensors and another matrix, as graphically illustrated in Figure 1(a). Afterwards, tensor ring (TR) decomposition [35, 23, 21], considered as a generalized form of TT decomposition, instead expands the near-border two matrices in TT factors to two third-order tensors and then constructs the cyclically sequential multi-linear products over $N$ third-order tensor factors, see Figure 1(b). Currently, a more sophisticated decomposition, referred fully-connected tensor network (FCTN) decomposition [37], has also been explored by establishing a structure evolved from the complete graph in graph theory, where all decomposed $N$th-order factors are connected to each other. As of now, TT, TR, and FCTN formats have been properly evaluated for their effectiveness and successfully applied to a variety of fields, e.g., classification [30], compressive sensing [5], and image/video recovery [8, 10, 36, 3, 28, 37, 12].

Despite the great success achieved by TT, TR, and FCTN decompositions, three intrinsic limitations are rather reflected in their TN structures. More specifically, (*i*) TT, even TR, models only establish the connection over adjacent latent factors while ignoring other possible interactions between non-adjacent dimensions, leading to an inadequate relation construction. (*ii*) From the perspective of tensor subspace, the intricate features inside an $N$th-order tensor can be well-preserved in the same dimensional tensor space without destroying the structures. However, when $N \geq 4$, TT and TR models express an $N$th-order tensor employing only third even lower-order factors, which definitely cannot form the desired space. (*iii*) From FCTN-ranks, FCTN model generates $(N^2 - N)/2$ hyper-parameters, which scale quadratically with tensor order $N$. Thus, numerically specifying an optimal collection of FCTN-ranks can be challenging for higher order, making it a costly and tedious task.

Given the above limitations, we argue that a more rational TN model should include three aspects. Namely, (*i*) all factors with dimension modes (i.e., $I_k$, $k = 1, 2, \cdots, N$) are certainly interconnected with their adjacent ones, while remaining as equivalently close as possible to the others. (*ii*) At least one $N$th-order factor is required to physically inherit the complex interactions from an $N$th-order tensor, thus expecting superior characterization for high-order tensors without sacrificing parameter storage. (*iii*) The TN structure keeps the scale of hyper-parameters lower, i.e., the smaller collection of ranks. Remarkably, the first two folds correspond to the characterization capabilities, which are crucial for TN models in high-order data recovery. Consequently, we propose the tensor wheel (TW) decomposition, which decomposes an $N$th-order tensor into both $N$ fourth-order ring factors and an $N$th-order core factor, and establishes the multi-linear products by wheel topology (see Figure 1(c)), aiming to agree above superior properties. The main ***contributions*** of this paper are as follows:

- We analytically propose a novel TW decomposition, which allows more expressive characterization for correlation than TT and TR formats, especially preserving the linear scaling for the number of hyper-parameters (TW-ranks) with increased tensor dimension (Section 2.3). Also, we present an alternating least squares (ALS)-based algorithm for learning TW decomposition (Section 2.4).

- We further provide one numerical application of TW decomposition, i.e., tensor completion (TC), and then formulate a TW decomposition-based TC model, abbreviated TW-TC. Moreover, we develop an efficient proximal alternating minimization (PAM)-based algorithm to solve the proposed TW-TC model and establish the theoretical proof of algorithm convergence (Section 3).

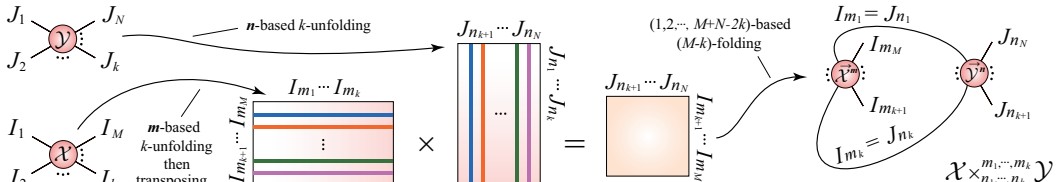

Figure 2: Graphical illustration of the generalized tensor $k$-contraction in Definition 3.

## 2 Tensor Wheel Decomposition

### 2.1 Notations and Preliminaries

In general, scalars, vectors, matrices, and tensors are denoted by lowercase letters, e.g., $x$, lowercase bold letters, e.g., $\mathbf{x}$, uppercase bold letters, e.g., $\mathbf{X}$, and calligraphic letters, e.g., $\mathcal{X}$, respectively. For an $N$th-order tensor $\mathcal{X} \in \mathbb{R}^{I_1 \times I_2 \times \cdots \times I_N}$, its element on index $(i_1, i_2, \cdots, i_N)$ is expressed by $\mathcal{X}(i_1, i_2, \cdots, i_N)$. The Frobenius norm is defined as $\|\mathcal{X}\|_F = \sqrt{\sum_{i_1, i_2, \cdots, i_N} \mathcal{X}(i_1, i_2, \cdots, i_N)^2}$. For $p, q \in \mathbb{N}^+$ and $p \leq q$, we especially denote by $\mathcal{X}_{p:q}$ the sequence $(\mathcal{X}_p, \cdots, \mathcal{X}_q)$ for brevity.

### 2.2 Basic Theory

To numerically construct the TW decomposition, we firstly develop several new tensor definitions.

**Definition 1 (Tensor Permutation).** *Given an $N$th-order tensor $\mathcal{X} \in \mathbb{R}^{I_1 \times I_2 \times \cdots \times I_N}$. Assuming that vector $\boldsymbol{n} = (n_1, n_2, \cdots, n_N)$ is a specific reordering of vector $(1, 2, \cdots, N)$, then we define the vector $\boldsymbol{n}$-based permutation of tensor $\mathcal{X}$ as a tensor $\vec{\mathcal{X}}^{\boldsymbol{n}} \in \mathbb{R}^{I_{n_1} \times I_{n_2} \times \cdots \times I_{n_N}}$, whose elements obey*

$$\vec{\mathcal{X}}^{\boldsymbol{n}}(i_{n_1}, i_{n_2}, \cdots, i_{n_N}) = \mathcal{X}(i_1, i_2, \cdots, i_N). \tag{1}$$

**Definition 2 (Generalized Tensor $k$-Unfolding).** *Let $\boldsymbol{n} = (n_1, n_2, \cdots, n_N)$ be a reordering of vector $(1, 2, \cdots, N)$, then for an $N$th-order tensor $\mathcal{X} \in \mathbb{R}^{I_1 \times I_2 \times \cdots \times I_N}$, the generalized $k$-unfolding $(0 \leq k \leq N, k \in \mathbb{Z})$ of $\mathcal{X}$ is defined as a matrix $\mathbf{X}_{[\boldsymbol{n};k]} \in \mathbb{R}^{\prod_{i=1}^k I_{n_i} \times \prod_{j=k+1}^N I_{n_j}}$, which requires*

$$\mathbf{X}_{[\boldsymbol{n};k]}(\overline{i_{n_1} i_{n_2} \cdots i_{n_k}}, \overline{i_{n_{k+1}} i_{n_{k+2}} \cdots i_{n_N}}) = \vec{\mathcal{X}}^{\boldsymbol{n}}(i_{n_1}, i_{n_2}, \cdots, i_{n_N}), \tag{2}$$

*where $\vec{\mathcal{X}}^{\boldsymbol{n}}$ is the tensor permutation of $\mathcal{X}$, the multi-indices $\overline{i_{n_1} i_{n_2} \cdots i_{n_k}}$ and $\overline{i_{n_{k+1}} i_{n_{k+2}} \cdots i_{n_N}}$ are defined by $1 + \sum_{i=1}^k (n_i - 1) \prod_{j=1}^{i-1} I_{n_j}$ and $1 + \sum_{i=k+1}^N (n_i - 1) \prod_{j=k+1}^{i-1} I_{n_j}$, respectively. When $k$ is $0$ and $N$, $\mathbf{x}_{[\boldsymbol{n};0]} \in \mathbb{R}^{1 \times \prod_{j=1}^N I_{n_j}}$ and $\mathbf{x}_{[\boldsymbol{n};N]} \in \mathbb{R}^{\prod_{i=1}^N I_{n_i} \times 1}$ imply two generalized vectorizations. Conversely, its inverse operator yields $\mathcal{X} = \text{Fold}_{[\boldsymbol{n};k]}(\mathbf{X}_{[\boldsymbol{n};k]})$ or $\mathcal{X} = \text{Fold}_{[\boldsymbol{n};k]}(\mathbf{x}_{[\boldsymbol{n};k]})$ for $k = 0, N$.*

Following Definition 2 with associated multi-indices, $\mathbf{X}_{[(k,1,2,\cdots,k-1,k+1,\cdots,N);1]}$ can be specialized as $\mathbf{X}_{(k)}$, which is the classical mode-$k$ unfolding in [17]. Also, $\mathbf{X}_{[(k,k+1,\cdots,N,1,2,\cdots,k-1);1]}$ can be specialized as $\mathbf{X}_{<k>}$ used in TR decomposition [35]. Correspondingly, the mode-$k$ folding operators of these two special unfoldings can be expressed as $\text{Fold}_{(k)}(\cdot)$ and $\text{Fold}_{<k>}(\cdot)$, respectively.

**Definition 3 (Generalized Tensor $k$-Contraction).** *Given an $M$th-order tensor $\mathcal{X} \in \mathbb{R}^{I_1 \times I_2 \times \cdots \times I_M}$ and an $N$th-order tensor $\mathcal{Y} \in \mathbb{R}^{J_1 \times J_2 \times \cdots \times J_N}$ with $k$ common modes $(1 \leq k \leq \min(M, N), k \in \mathbb{Z})$. Assume that two vectors $\boldsymbol{m} = (m_1, m_2, \cdots, m_M)$ and $\boldsymbol{n} = (n_1, n_2, \cdots, n_N)$ respectively indicate the reordering of vectors $(1, 2, \cdots, M)$ and $(1, 2, \cdots, N)$, satisfying $I_{m_i} = J_{n_i}$ for $i = 1, 2, \cdots, k$, $m_{k+1} < m_{k+2} < \cdots < m_M$ and $n_{k+1} < n_{k+2} < \cdots < n_N$. Then the generalized $k$-contraction between $\mathcal{X}$ and $\mathcal{Y}$ along the $k$ modes specifies an $(M + N - 2k)$th-order tensor $\mathcal{X} \times_{n_1, n_2, \cdots, n_k}^{m_1, m_2, \cdots, m_k} \mathcal{Y} \in \mathbb{R}^{I_{m_{k+1}} \times \cdots \times I_{m_M} \times J_{n_{k+1}} \times \cdots \times J_{n_N}}$, which is given (also intuitively shown in Figure 2) as follows,*

$$\mathcal{X} \times_{n_1, n_2, \cdots, n_k}^{m_1, m_2, \cdots, m_k} \mathcal{Y} = \text{Fold}_{[(1,2,\cdots,M+N-2k);M-k]}(\mathbf{X}_{[\boldsymbol{m};k]}^T \mathbf{Y}_{[\boldsymbol{n};k]}), \tag{3}$$

*where $\mathbf{X}_{[\boldsymbol{m};k]}$ and $\mathbf{Y}_{[\boldsymbol{n};k]}$ are the $\boldsymbol{m}$-based and $\boldsymbol{n}$-based $k$-unfoldings of tensors $\mathcal{X}$ and $\mathcal{Y}$, respectively. Significantly, the generalized tensor $k$-contraction operator requires a left-side precedence and non-commutative property. That is, neither the associative nor commutative properties are satisfied.*

## 2.3 Tensor Wheel Model

As illustrated in Figure 1(c), let $\mathcal{X} \in \mathbb{R}^{I_1 \times I_2 \times \cdots \times I_N}$ be an $N$th-order tensor, then TW decomposition aims to parameterize it by both $N$ fourth-order ring factors $\mathcal{G}_k \in \mathbb{R}^{R_k \times I_k \times L_k \times R_{k+1}}$, $k = 1, 2, \cdots, N$, and an $N$th-order core factor $\mathcal{C} \in \mathbb{R}^{L_1 \times L_2 \times \cdots \times L_k \times \cdots \times L_N}$. When $k = N$, $R_{N+1}$ stands for $R_1$. Mathematically, TW model establishes the element-wise relation as follows,

$$\mathcal{X}(i_1, i_2, \cdots, i_N) = \sum_{r_1=1}^{R_1} \sum_{r_2=1}^{R_2} \cdots \sum_{r_N=1}^{R_N} \sum_{l_1=1}^{L_1} \cdots \sum_{l_N=1}^{L_N} \{\mathcal{G}_1(r_1, i_1, l_1, r_2)\mathcal{G}_2(r_2, i_2, l_2, r_3) \cdots \tag{4}$$
$$\mathcal{G}_k(r_k, i_k, l_k, r_{k+1}) \cdots \mathcal{G}_N(r_N, i_N, l_N, r_1)\mathcal{C}(l_1, l_2, \cdots, l_N)\}.$$

According to the wheel topology, when the tensor contractions are executed in the simplest manner, TW model (4) can be naturally expressed in the tensor form by $\mathcal{X} = \mathcal{G}_1 \times_1^4 \mathcal{G}_2 \times_1^6 \cdots \times_1^{2k} \mathcal{G}_k \times_1^{2k+2} \cdots \times_{1,4}^{2N,1} \mathcal{G}_N \times_{1,2,\cdots,N}^{2,4,\cdots,2N} \mathcal{C}$, which provides the actual numerical computation for TW decomposition. Symbolically, we employ $TW[\![\mathcal{G}_1, \mathcal{G}_2, \cdots, \mathcal{G}_N; \mathcal{C}]\!]$ or, more compactly, $TW[\![\{\mathcal{G}_k\}_{k=1}^N; \mathcal{C}]\!]$ to denote TW decomposition. Moreover, the ring factors $\mathcal{G}_k$, $k = 1, 2, \cdots, N$, and core factor $\mathcal{C}$ are collectively called *TW factors*. Apart from the above basic form, TW decomposition also has numerous generalized tensor forms, but enumerating all the cases is laborious and pointless. In what follows, we exhibit two special tensor forms by Theorems 1 and 2, aiming to illustrate its invariance property.

**Theorem 1** (**Core-Centered Circular Invariance**). *Given an $N$th-order tensor $\mathcal{X} \in \mathbb{R}^{I_1 \times I_2 \times \cdots \times I_N}$ and its TW decomposition $TW[\![\{\mathcal{G}_k\}_{k=1}^N; \mathcal{C}]\!]$. Assume that $\boldsymbol{n} = (n_1, n_2, \cdots, n_N)$ is the circular reordering of vector $(1, 2, \cdots, N)$, then the core-centered invariance gives $\vec{\mathcal{X}}^{\boldsymbol{n}} = TW[\![\{\mathcal{G}_{n_k}\}_{k=1}^N; \vec{\mathcal{C}}^{\boldsymbol{n}}]\!]$.*

**Theorem 2** (**Core-Connected Invariance**). *Assume that the TW decomposition of $\mathcal{X} \in \mathbb{R}^{I_1 \times \cdots \times I_N}$ is $\vec{\mathcal{X}}^{\boldsymbol{n}} = TW[\![\{\mathcal{G}_{n_k}\}_{k=1}^N; \vec{\mathcal{C}}^{\boldsymbol{n}}]\!]$, where $\boldsymbol{n} = (n_1, n_2, \cdots, n_N)$ is any vector that circularly shifts vector $(1, 2, \cdots, N)$. Let vector $\boldsymbol{e} = (n_1, n_k, n_2, \cdots, n_{k-1}, n_{k+1}, \cdots, n_N)$ $(3 \leq k < N, k \in \mathbb{Z})$, then*

$$\vec{\mathcal{X}}^{\boldsymbol{e}} = (\mathcal{G}_{n_1} \times_1^3 \vec{\mathcal{C}}^{\boldsymbol{e}} \times_3^4 \mathcal{G}_{n_k}) \times_{1,3,\cdots,2k-3,2k-2}^{3,4,\cdots,k+1,N+2} \mathcal{U}_{n_2,\cdots,n_{k-1}} \times_{2(N-k)+2,3,5,\cdots,2(N-k)+1,1}^{1,3,4,\cdots,N-k+2,N-k+4} \mathcal{V}_{n_{k+1},\cdots,n_N},$$

*where $\mathcal{U}_{n_2,\cdots,n_{k-1}} = \mathcal{G}_{n_2} \times_1^4 \cdots \times_1^{2(k-2)} \mathcal{G}_{n_{k-1}}$ and $\mathcal{V}_{n_{k+1},\cdots,n_N} = \mathcal{G}_{n_{k+1}} \times_1^4 \cdots \times_1^{2(N-k)} \mathcal{G}_{n_N}$.*

Theorem 2 reveals any two non-adjacent ring factors with dimension modes (i.e., $I_k$, $k = 1, 2, \cdots, N$) can be connected by the core factor. Accordingly, TW topology can comprehensively establish all possible mode interactions of a high-order tensor without being impeded by other ring factors, which endows TW decomposition with adequate relation construction over TT and TR decompositions.

**Theorem 3** (**Tensor Subwheel Equation**). *Assume that the TW decomposition of $\mathcal{X} \in \mathbb{R}^{I_1 \times I_2 \times \cdots \times I_N}$ is $\vec{\mathcal{X}}^{\boldsymbol{n}} = TW[\![\{\mathcal{G}_{n_k}\}_{k=1}^N; \vec{\mathcal{C}}^{\boldsymbol{n}}]\!]$, where $\boldsymbol{n} = (n_1, n_2, \cdots, n_N)$ is any vector that circularly shifts vector $(1, 2, \cdots, N)$. Let $\boldsymbol{m} = (N+1, N+2, 1, 2, \cdots, N)$ and $\boldsymbol{v} = (2, 4, \cdots, 2N, 1, 3, \cdots, 2N-1)$, then there inherently exists the following two tensor subwheel equations,*

$$\mathbf{X}_{<n_N>} = (\mathbf{G}_{n_N})_{(2)}(\mathbf{M}_{\neq n_N})_{[\boldsymbol{m};3]} \quad and \quad \vec{\mathbf{x}}_{[1:N,0]}^{\boldsymbol{n}} = \vec{\mathbf{c}}_{[1:N,0]}^{\boldsymbol{n}}(\mathbf{N}_{\neq \mathcal{C}})_{[\boldsymbol{v};N]}, \tag{5}$$

*where $\mathcal{M}_{\neq n_N} \in \mathbb{R}^{R_{n_1} \times I_{n_1} \times \cdots \times I_{n_{N-1}} \times R_{n_N} \times L_{n_N}}$ is an $(N+2)$th-order subwheel tensor, which merges all TW factors but $\mathcal{G}_{n_N}$, i.e., $\mathcal{M}_{\neq n_N} = \mathcal{G}_{n_1} \times_1^4 \cdots \times_1^{2k} \mathcal{G}_{n_k} \times_1^{2k+2} \cdots \times_1^{2N-2} \mathcal{G}_{n_{N-1}} \times_{1,2,\cdots,N-1}^{3,5,\cdots,2N-1} \vec{\mathcal{C}}^{\boldsymbol{n}}$, and $\mathcal{N}_{\neq \mathcal{C}} \in \mathbb{R}^{I_{n_1} \times L_{n_1} \times \cdots \times I_{n_N} \times L_{n_N}}$ is another $2N$th-order subwheel tensor obtained by only merging $\{\mathcal{G}_{n_k}\}_{k=1}^N$, i.e., $\mathcal{N}_{\neq \mathcal{C}} = \mathcal{G}_{n_1} \times_1^4 \cdots \times_1^{2k} \mathcal{G}_{n_k} \times_1^{2k+2} \cdots \times_{1,4}^{2N,1} \mathcal{G}_{n_N}$.*

Theorem 3 clarifies that all TW factors, i.e., $\{\mathcal{G}_k\}_{k=1}^N$ and $\mathcal{C}$, can be individually separated from $TW[\![\{\mathcal{G}_k\}_{k=1}^N; \mathcal{C}]\!]$, which contributes to those alternating direction-based algorithms, e.g., ALS and PAM ones in Sections 2.4 and 3.1, respectively. Furthermore, $\vec{\mathbf{x}}_{[1:N,0]}^{\boldsymbol{n}} = \vec{\mathbf{c}}_{[1:N,0]}^{\boldsymbol{n}}(\mathbf{N}_{\neq \mathcal{C}})_{[\boldsymbol{v};N]}$ in formula (5) is essentially $\vec{\mathcal{X}}^{\boldsymbol{n}} = \vec{\mathcal{C}}^{\boldsymbol{n}} \times_{2,4,\cdots,2N}^{1,2,\cdots,N} \mathcal{N}_{\neq \mathcal{C}}$, appearing as a tensor subspace representation. Since all modes of $\vec{\mathcal{X}}^{\boldsymbol{n}}$ are contained in $\mathcal{N}_{\neq \mathcal{C}}$, $\vec{\mathcal{C}}^{\boldsymbol{n}}$ is purely a coefficient tensor, and $\mathcal{N}_{\neq \mathcal{C}}$ can be regarded as a basis tensor. From such a perspective, the intrinsic structure of a high-order tensor can be well-preserved by the core factor $\mathcal{C}$, leading to a high characterization ability of TW decomposition.

**Theorem 4.** *Assume that $\mathcal{X} = TW[\![\{\mathcal{G}_k\}_{k=1}^N; \mathcal{C}]\!]$ with $N$ ring factors $\mathcal{G}_k \in \mathbb{R}^{R_k \times I_k \times L_k \times R_{k+1}}$, then*

$$\mathrm{Rank}(\mathbf{X}_{(k)}) = \mathrm{Rank}(\mathbf{X}_{<k>}) \leq L_k \prod_{i=k}^{k+1} R_i, \quad k = 1, 2, \cdots, N. \tag{6}$$

---

**Algorithm 1** The Alternating Least Squares Algorithm for TW Decomposition (TW-ALS).

---

**Input:** An $N$th-order tensor $\mathcal{X} \in \mathbb{R}^{I_1 \times I_2 \times \cdots \times I_N}$, the manually given TW-ranks $\mathbf{r}$, and threshold $\epsilon$.
**Initialization:** The randomized $\mathcal{G}_k \in \mathbb{R}^{R_k \times I_k \times L_k \times R_{k+1}}$, $k = 1, \cdots, N$, and $\mathcal{C} \in \mathbb{R}^{L_1 \times L_2 \times \cdots \times L_N}$.

1: **while** *not converged* **do**
2:     Record the last-update result $\mathcal{T}_{last} \leftarrow \text{TW}[\![\{\mathcal{G}_k\}_{k=1}^{N}; \mathcal{C}]\!]$.
3:     **for** $k = 1$ to $N$ **do**
4:         Obtain the subwheel tensor $\mathcal{M}_{\neq k}$ and vector $\boldsymbol{m}$ by Theorem 3.
5:         $(\mathbf{G}_k)_{(2)} \leftarrow \underset{(\mathbf{G}_k)_{(2)}}{\arg \min} \|\mathbf{X}_{<k>} - (\mathbf{G}_k)_{(2)}(\mathbf{M}_{\neq k})_{[\boldsymbol{m};3]}\|_F$.
6:     **end for**
7:     Obtain the subwheel tensor $\mathcal{N}_{\neq \mathcal{C}}$ and vector $\boldsymbol{v}$ by Theorem 3.
8:     $\mathbf{c}_{[1:N,0]} \leftarrow \underset{\mathbf{c}_{[1:N,0]}}{\arg \min} \|\mathbf{x}_{[1:N,0]} - \mathbf{c}_{[1:N,0]}(\mathbf{N}_{\neq \mathcal{C}})_{[\boldsymbol{v};N]}\|_F$.
9:     Check the convergence criterion: $\|\text{TW}[\![\{\mathcal{G}_k\}_{k=1}^{N}; \mathcal{C}]\!] - \mathcal{T}_{last}\|_F / \|\mathcal{T}_{last}\|_F < \epsilon$.
10: **end while**
**Output:** TW factors $\mathcal{G}_k$, $k = 1, 2, \cdots, N$, and $\mathcal{C}$.

---

Theorem 4 justifies that scalars $R_k$ and $L_k$, $k = 1, 2, \cdots, N$, can upper bound the rank of all dimension-mode tensor unfolding and control the low-rankness behavior to a certain extent. Thus, they are assigned as the *TW-ranks*, usually collected by a vector $\mathbf{r} = (R_1, \cdots, R_N, L_1, \cdots, L_N) \in \mathbb{R}^{2N}$. The TW-ranks essentially determines the actual TN structure, e.g., Tucker topology when $R_k = 1$ and TR topology when $L_k = 1$ with $k = 1, 2, \cdots, N$, graphically proving the superiority of TW decomposition over both Tucker and TR decompositions. Along such a perspective, TT, TR, and FCTN decompositions can be viewed as a special case of the adaptive TN (ATN) decomposition [11], which adaptively identifies TN structures by eliminating the relevant edges from the complete graph topology. Nevertheless, provided that decomposing the same $N$th-order tensor, TW decomposition is exactly excluded from the setting of ATN decomposition since the latter cannot fit the structures with an internal core tensor, e.g., Tucker [26], hierarchical Tucker [9], and projected entangled state pairs (PEPS) [27] topologies. Compared with FCTN decomposition with $(N^2 - N)/2$ ranks, TW decomposition forms the collection of $2N$ ranks, which grows only linearly rather than quadratically with the tensor order $N$, thereby improving its flexibility for higher-order applications.

**Limitation.** The proposed TW topology allows for more expressive characterization capability using the linear number of hyper-parameters. Nonetheless, such a structure inevitably leads to a limitation on the storage costs. For a given $N$th-order tensor $\mathcal{X} \in \mathbb{R}^{I \times \cdots \times I}$, whose TW-ranks is assumed as the same value $R$, i.e., $\mathbf{r} = (R, \cdots, R)$, then TW decomposition requires $\mathcal{O}(NIR^3 + R^N)$ parameter costs for storage. It is clear that the $\mathcal{O}(NIR^3 + R^N)$ scales exponentially with the tensor order $N$, resulting in a greater storage burden for larger-scale and higher-order tensor recovery problems.

Unlike Tucker decomposition, TW one considers the potential relationship between adjacent factors and establishes a connection for a higher characterization capacity. Actually, such a strategy also reduces the loadings of core factor, which contributes a smaller $\{L_i\}_{i=1}^{N}$ in TW-ranks than Tucker decomposition, thus alleviating the limitations of high storage and computational complexity.

## 2.4 Alternating Least Squares (ALS) Algorithm

In this section, we develop an ALS-based algorithm to perform TW decomposition, called TW-ALS. Given $\mathcal{X} \in \mathbb{R}^{I_1 \times I_2 \times \cdots \times I_N}$ and its predefined TW-ranks, then the TW-ALS algorithm mainly aims to seek a set of TW factors such that a relative error between $\mathcal{X}$ and $\text{TW}[\![\{\mathcal{G}_k\}_{k=1}^{N}; \mathcal{C}]\!]$ is minimum, i.e.,

$$\min_{\mathcal{G}_1, \mathcal{G}_2, \cdots, \mathcal{G}_N, \mathcal{C}} \|\mathcal{X} - \text{TW}[\![\{\mathcal{G}_k\}_{k=1}^{N}; \mathcal{C}]\!]\|_F. \tag{7}$$

Based on Theorem 3, minimization problem (7) can be optimized under the ALS framework by alternatively solving $N + 1$ least-squares subproblems, i.e., $\min_{(\mathbf{G}_k)_{(2)}} \|\mathbf{X}_{<k>} - (\mathbf{G}_k)_{(2)}(\mathbf{M}_{\neq k})_{[\boldsymbol{m};3]}\|_F$, $k = 1, 2, \cdots, N$, and $\min_{\mathbf{c}_{[1:N,0]}} \|\mathbf{x}_{[1:N,0]} - \mathbf{c}_{[1:N,0]}(\mathbf{N}_{\neq \mathcal{C}})_{[\boldsymbol{v};N]}\|_F$. Compared to formula (5), since we utilize the simplest case without permutation for updating $\mathcal{C}$, $\mathbf{x}_{[1:N,0]}$ and $\mathbf{c}_{[1:N,0]}$ imply $\vec{\mathbf{x}}_{[1:N,0]}^{\boldsymbol{n}}$ and $\vec{\mathbf{c}}_{[1:N,0]}^{\boldsymbol{n}}$ with $\boldsymbol{n} = (1, 2, \cdots, N)$, respectively. The numerical procedure is detailed in Algorithm 1.

**Algorithm 2** The Proximal Alternating Minimization (PAM)-Based Solver for TW-TC Model.

---

**Input:** The observed tensor $\mathcal{F} \in \mathbb{R}^{I_1 \times I_2 \times \cdots \times I_N}$, the location set $\Omega$, the TW-ranks $\mathbf{r}$, the step size $s = 20$, $\rho = 0.001$, $t_{\max} = 1000$, $\Omega^c$ indicating the complementary set of $\Omega$, and $\epsilon = 10^{-5}$.

**Initialization:** The iteration step $t = 0$, $\mathcal{X}^{(0)} = \mathcal{F}$, and the randomized $\mathcal{G}_k^{(0)} \in \mathbb{R}^{R_k \times I_k \times L_k \times R_{k+1}}$, $k = 1, 2, \cdots, N$, and $\mathcal{C}^{(0)} \in \mathbb{R}^{L_1 \times L_2 \times \cdots \times L_N}$.

1: **while** *not converged* and $t < t_{\max}$ **do**
2:      Determine the subwheel tensor $\mathcal{M}_{\neq k}^{(t)}$ and vector $\boldsymbol{m}$ relying upon Theorem 3.
3:      $\mathcal{G}_k^{(t+1)} = \texttt{Fold}_{(2)}\big\{ \big(\mathbf{X}_{<k>}^{(t)}(\mathbf{M}_{\neq k}^{(t)})_{[\boldsymbol{m};3]}^T + \rho(\mathbf{G}_k^{(t)})_{(2)}\big) / \big((\mathbf{M}_{\neq k}^{(t)})_{[\boldsymbol{m};3]}(\mathbf{M}_{\neq k}^{(t)})_{[\boldsymbol{m};3]}^T + \rho\mathbf{I}\big) \big\}$.
4:      Determine the subwheel tensor $\mathcal{N}_{\neq\mathcal{C}}^{(t)}$ and vector $\boldsymbol{v}$ relying upon Theorem 3.
5:      **if** $t > 200$ and $\texttt{mod}(t, s) \neq 0$ (i.e., $t$ modulo $s$ is not 0) **then**
6:          $\mathcal{C}^{(t+1)} = \mathcal{C}^{(t)}$.
7:      **else**
8:          $\mathcal{C}^{(t+1)} = \texttt{Fold}_{[1:N,0]}\{(\mathbf{x}_{[1:N,0]}^{(t)}(\mathbf{N}_{\neq\mathcal{C}}^{(t)})_{[\boldsymbol{v};N]}^T + \rho\mathbf{c}_{[1:N,0]}^{(t)})/((\mathbf{N}_{\neq\mathcal{C}}^{(t)})_{[\boldsymbol{v};N]}(\mathbf{N}_{\neq\mathcal{C}}^{(t)})_{[\boldsymbol{v};N]}^T + \rho\mathbf{I})\}$.
9:      **end if**
10:     $\mathcal{X}^{(t+1)} = \mathcal{P}_{\Omega^c}\big\{(\text{TW}[\![\{\mathcal{G}_k^{(t+1)}\}_{k=1}^N; \mathcal{C}^{(t+1)}]\!] + \rho\mathcal{X}^{(t)})/(1 + \rho)\big\} + \mathcal{P}_{\Omega}(\mathcal{F})$.
11:     Check the convergence criterion: $\|\mathcal{X}^{(t+1)} - \mathcal{X}^{(t)}\|_F / \|\mathcal{X}^{(t)}\|_F < \epsilon$.
12:     $t \leftarrow t + 1$.
13: **end while**
**Output:** The recovered tensor $\mathcal{X} \in \mathbb{R}^{I_1 \times I_2 \times \cdots \times I_N}$.

---

## 3 Numerical Application to Tensor Completion

To validate the rationality and superiority of TW decomposition, we employ it to one classical tensor recovery problem, i.e., tensor completion (TC), whose objective is recovering the missing entries from a partially observed tensor, then formulating a TW decomposition-based TC (TW-TC) model. The proposed TW-TC method is committed to exploring latent TW factors from the partially observed tensor, then predicting the missing entries using these constructed factors. This essentially evaluates the potential capability of TW decomposition to characterize the high-order tensors.

### 3.1 Model and Algorithm

Given a partially observed tensor $\mathcal{F} \in \mathbb{R}^{I_1 \times I_2 \times \cdots \times I_N}$ at location set $\Omega$, then the proposed TW-TC model aims to identify an optimum-TW-ranks approximation $\mathcal{X} \in \mathbb{R}^{I_1 \times I_2 \times \cdots \times I_N}$ of the tensor $\mathcal{F}$. Mathematically, the TW-TC model can be formulated as the unconstrained problem as follows,

$$\min_{\mathcal{X}, \mathcal{G}_{1:N}, \mathcal{C}} \frac{1}{2}\|\mathcal{X} - \text{TW}[\![\{\mathcal{G}_k\}_{k=1}^N; \mathcal{C}]\!]\|_F^2 + \iota(\mathcal{X}) \text{ with } \iota(\mathcal{X}) := \begin{cases} 0, & \mathcal{X} \in \{\mathcal{L}: \mathcal{P}_{\Omega}(\mathcal{L}) = \mathcal{P}_{\Omega}(\mathcal{F})\}; \\ \infty, & \text{otherwise}, \end{cases}$$
$$(8)$$

where $\mathcal{P}_{\Omega}(\cdot)$ is a projection function keeping the entries in $\Omega$ while forcing all the others to zeros.

Obviously, the object function in formula (8) is not jointly but independently convex for all variables. Thus, we develop a proximal alternating minimization (PAM) [2] based algorithm to alternately and recursively optimize each variable, leading to the following procedure for $k = 1, 2, \cdots, N$,

$$\begin{cases} \mathcal{G}_k^{(t+1)} \in \arg\min_{\mathcal{G}_k} \left\{ \frac{1}{2}\|\mathcal{X}^{(t)} - \text{TW}[\![\mathcal{G}_{1:k-1}^{(t+1)}, \mathcal{G}_k, \mathcal{G}_{k+1:N}^{(t)}; \mathcal{C}^{(t)}]\!]\|_F^2 + \frac{\rho}{2}\|\mathcal{G}_k - \mathcal{G}_k^{(t)}\|_F^2 \right\}, \\ \mathcal{C}^{(t+1)} \in \arg\min_{\mathcal{C}} \left\{ \frac{1}{2}\|\mathcal{X}^{(t)} - \text{TW}[\![\mathcal{G}_{1:N}^{(t+1)}; \mathcal{C}]\!]\|_F^2 + \frac{\rho}{2}\|\mathcal{C} - \mathcal{C}^{(t)}\|_F^2 \right\}, \\ \mathcal{X}^{(t+1)} \in \arg\min_{\mathcal{X}} \left\{ \frac{1}{2}\|\mathcal{X} - \text{TW}[\![\mathcal{G}_{1:N}^{(t+1)}; \mathcal{C}^{(t+1)}]\!]\|_F^2 + \frac{\rho}{2}\|\mathcal{X} - \mathcal{X}^{(t)}\|_F^2 + \iota(\mathcal{X}) \right\}, \end{cases} \quad (9)$$

where $\rho > 0$ is a proximal parameter. Following Theorem 3 again, each univariate minimization problem in formula (9) can certainly be reduced to a linear least-squares problem, which is expressed by matrix form in the supplementary material, as in Section 2.4. Then, the corresponding closed-form solutions can easily be given. In most numerical TC experiments, the variable $\mathcal{C}$ typically tends to be

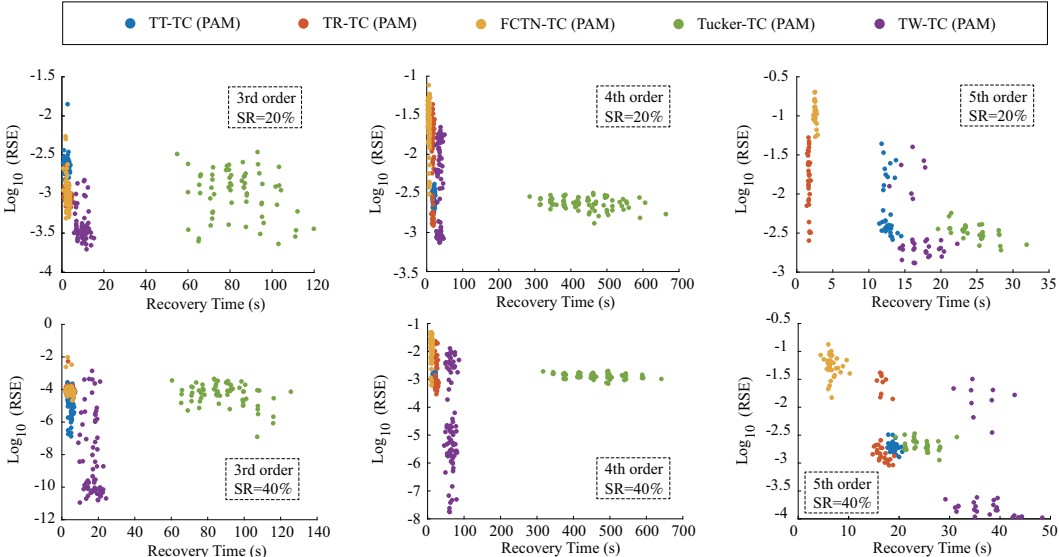

Figure 3: Recovery performance (i.e., $\text{Log}_{10}$(RSE)) against recovery time. The TT-TC (PAM), TR-TC (PAM), FCTN-TC (PAM), Tucker-TC (PAM), and proposed TW-TC algorithms are evaluated on synthetic data under six different cases. The first column: 64 third-order tensor experiments with SR=20% and 40%; The second column: 81 fourth-order tensor experiments with SR=20% and 40%; The third column: 32 fifth-order tensor experiments with SR=20% and 40%. The smaller $\text{Log}_{10}$(RSE) value implies higher recovery accuracy.

stable when it exceeds 200 iterations. Therefore, we utilize the modulo operation (marked $\text{mod}(\cdot,\cdot)$) with a step size to intermittently update $\mathcal{C}$ after 200 iterations, aiming to save computing resources. The PAM-based solving procedure for the developed TW-TC model is summarized in Algorithm 2.

## 3.2 Computational Complexity Analysis

For simplicity, we assume that all dimensions of an $N$th-order tensor $\mathcal{X} \in \mathbb{R}^{I_1 \times \cdots \times I_k \times \cdots \times I_N}$ are $I$ and all values of TW-ranks are $R$, i.e., $I_k = I$ and $R_k = L_k = R$ for $k = 1, 2, \cdots, N$. Then, the computational complexities involved in updating factors $\{\mathcal{G}_k\}_{k=1}^{N}$, $\mathcal{C}$, and $\mathcal{X}$ are $\mathcal{O}(N \sum_{l=2}^{N-1} I^l R^{l+3} + NI^N R^3 + NR^9)$, $\mathcal{O}(I^N R^{2N})$, and $\mathcal{O}(I^N R^N)$, respectively. Therefore, the computational complexity for each nonintermittent iteration (i.e., $t \in \{t : 0 \leq t \leq 200 \text{ or } \text{mod}(t, s) = 0\}$) in Algorithm 2 is $\mathcal{O}(N \sum_{l=2}^{N-1} I^l R^{l+3} + NI^N R^3 + I^N R^{2N})$. Akin to the storage parameters in the **Limitation** of Section 2.3, the computational complexity also increases exponentially with dimensionality $N$, implying that more computing resources are required for higher-order tensor recovery tasks.

## 3.3 Convergence Analysis

This section consists in theoretically proving the convergence of Algorithm 2. Before that, we employ the $\mathcal{Z} = (\mathcal{X}, \mathcal{G}_{1:N}, \mathcal{C})$ and $\Phi(\mathcal{Z})$ to denote the variables and objective function in (8), respectively.

**Theorem 5.** *Let $\{\mathcal{Z}^{(t)}\}_{t \in \mathbb{N}}$ be the sequence generated by Algorithm 2, then it globally converges to a critical point (i.e., local minimum point) $\mathcal{Z}^* = (\mathcal{X}^*, \mathcal{G}_{1:N}^*, \mathcal{C}^*)$ of the objective function $\Phi(\mathcal{Z})$.*

*Proof.* According to the finite length property (see [4, Theorem 1, pag 479]), the sequence $\{\mathcal{Z}^{(t)}\}_{t \in \mathbb{N}}$ can converge to the critical point $\mathcal{Z}^*$ of $\Phi(\mathcal{Z})$, since the following Lemmas 1-4 are satisfied. □

**Lemma 1.** *The objective function $\Phi(\mathcal{Z})$ in formula (8) is a Kurdyka-Łojasiewicz (KŁ) function.*

**Lemma 2 (Sufficient Decrease Condition).** *Let $\{\mathcal{Z}^{(t)}\}_{t \in \mathbb{N}}$ be the sequence generated by Algorithm 2. Then, the sequence $\{\Phi(\mathcal{Z}^{(t)})\}_{t \in \mathbb{N}}$ explicitly satisfies $\Phi(\mathcal{Z}^{(t)}) - \Phi(\mathcal{Z}^{(t+1)}) \geq \rho/2\|\mathcal{Z}^{(t+1)} - \mathcal{Z}^{(t)}\|_F^2$, where $\|\mathcal{Z}^{(t+1)} - \mathcal{Z}^{(t)}\|_F^2 = \|\mathcal{X}^{(t+1)} - \mathcal{X}^{(t)}\|_F^2 + \sum_{k=1}^{N} \|\mathcal{G}_k^{(t+1)} - \mathcal{G}_k^{(t)}\|_F^2 + \|\mathcal{C}^{(t+1)} - \mathcal{C}^{(t)}\|_F^2$.*

**Lemma 3 (Relative Error Condition).** *Let $\{\mathcal{Z}^{(t)}\}_{t \in \mathbb{N}}$ be the sequence generated by Algorithm 2. Then, there exists $\|\partial\Phi(\mathcal{Z}^{(t+1)})\|_F \leq \{L_\Phi + (N+2)\rho\}\|\mathcal{Z}^{(t+1)} - \mathcal{Z}^{(t)}\|_F$, where $L_\Phi$ sums the Lipschitz constants of $\{\partial_{\mathcal{G}_k}\Phi(\mathcal{Z})\}_{k=1}^{N}$ and $\partial_{\mathcal{C}}\Phi(\mathcal{Z})$, i.e., $L_\Phi = \sum_{k=1}^{N} L_{\mathcal{G}_k} + L_{\mathcal{C}}$.*

**Lemma 4.** *Let $\{\mathcal{Z}^{(t)}\}_{t\in\mathbb{N}}$ be the sequence generated by Algorithm 2, then it is bounded.*

Considering the page limitations, the proof of Lemmas 1-4 is detailed in the supplementary material.

### 3.4 Numerical Experiments

In this section, we design substantial numerical experiments on synthetic and real-world data to verify the superiority of the proposed TW-TC method compared with others, which are constructed based on several commonly used tensor decompositions. All the experiments are implemented in MATLAB (R2021a) on a computer of 64Gb RAM and Intel(R) Core(TM) i9-10900KF CPU: @3.70 GHz.

#### 3.4.1 Synthetic Data Completion

To testify the superiority of TW decomposition over four related decompositions, i.e., TT [22], TR [35], FCTN [37], and Tucker [26] ones, we firstly conduct the numerical experiments on synthetic tensor data by horizontally comparing the TW-TC method with their PAM-based TC algorithms, i.e., TT-TC (PAM), TR-TC (PAM), FCTN-TC (PAM), and Tucker-TC (PAM), respectively. Since the Tucker factors comprise an underlying high-order structure, which may be closer to reality, we construct the synthetic tensors by Tucker decomposition using Tucker factors sampled from the uniform distribution $U(0,1)$. More specifically, the synthetic data consists of 64 third-order, 81 fourth-order, and 32 fifth-order tensors, whose sizes are $\{I_1 \times I_2 \times I_3 : I_1, I_2, I_3 \in \{45, 50, 55, 60\}\}$, $\{I_1 \times I_2 \times I_3 \times I_4 : I_1, I_2, I_3, I_4 \in \{18, 20, 22\}\}$, and $\{I_1 \times I_2 \times I_3 \times I_4 \times I_5 : I_1, I_2, I_3, I_4, I_5 \in \{7, 8\}\}$, and Tucker-ranks are $(6, 6, 6)$, $(5, 5, 5, 5)$, and $(3, 3, 3, 3, 3)$, respectively. All synthetic data are numerically renormalized into [0, 1]. Subsequently, the partially observed tensors are generated by random sampling with two sampling rates (SRs): $20\%$, $40\%$. For each case, the hyper-parameters, i.e., TT-ranks, TR-ranks, FCTN-ranks, Tucker-ranks, and TW-ranks, also the storage complexities are provided in Appendix C. The performance is assessed by the residual standard error (RSE), i.e., $\|\mathcal{X} - \mathcal{X}_{real}\|_F / \|\mathcal{X}_{real}\|_F$, where $\mathcal{X}$ is the recovered tensor.

Figure 3 depicts the recovery result versus recovery time for five PAM-based TC models under six different cases. Compared with TT-TC (PAM), TR-TC (PAM), and FCTN-TC (PAM), the TW-TC method achieves the optimal RSE values for the majority of samples using both competitive computing efficiency and lower storage complexity (see Appendix C), which experimentally demonstrates the superior characterization ability of TW decomposition and strongly supports the theoretical analysis presented in Sections 1 and 2.3. Moreover, the faster computational efficiency of TW-TC over the Tucker-TC model implies that TW decomposition can alleviate the curse of high-order core factor on computational burden, achieving greater applicability than Tucker decomposition.

#### 3.4.2 Real-World Data Completion

Furthermore, we investigate the TW-TC method on real-world data experiments by comparing it with several tensor decomposition-based state-of-the-art methods, including HaLRTC [18], t-SVD [33], TMacTT [3], TRLRF [32], and FCTN-TC [37]. All hyper-parameters adopted in these competitors are fine-tuned within a specific range suggested by their authors to achieve better performance. The real-world data mainly comprises two types, i.e., the multispectral image (MSI) and the video data. Similarly, all experimental data are numerically pre-normalized into [0, 1]. Afterwards, the partially observed tensors are created by random sampling with three SRs: $5\%$, $10\%$, $20\%$. A popular metric, i.e., the peak signal-to-noise ratio (PSNR), is employed for quantitative evaluation. When applied to the MSI and video data, the mean of PSNRs across all channels is measured, termed MPSNR.

**MSI Data.** The tested MSI data sizes $200 \times 200 \times 31$ (i.e., height $\times$ width $\times$ spectral), called *Toy*, which is cropped from the CAVE dataset[2]. Regarding TW-ranks, we empirically assign $R_1 = L_3$ and $R_3 = L_1 = L_2$ based on observations of multiple third-order real-world data experiments, and then select ranks $R_1$, $R_2$ and $R_3$ from the candidate sets $\{3, 4, 5\}$, $\{10, 15, 20, 25\}$ and $\{2, 3\}$, respectively. Numerically, Table 1 (Data Index: *Toy*) reports the MPSNR values under three different SRs and the average computing costs of all compared methods. Also, Figure 4 presents their visual performances and the corresponding error maps when SR=$20\%$. From the quantitative and qualitative results, we observe that our TW-TC model achieves higher MPSNR values and better visual reconstruction than the other methods, confirming its effectiveness for real-world third-order data.

---

[2]https://www.cs.columbia.edu/CAVE/databases/multispectral/

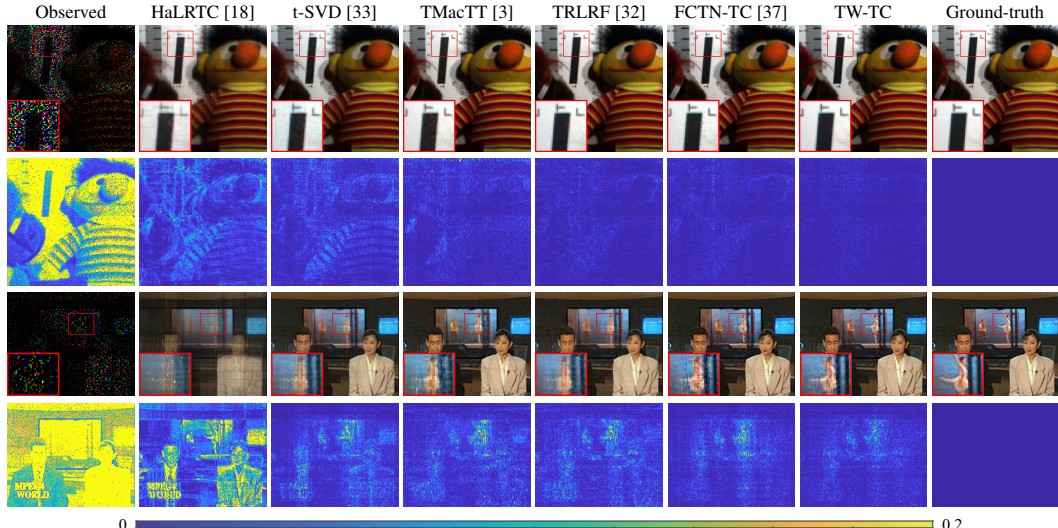

Figure 4: Visualization of the recovered results on MSI *Toy* (SR=20%) and CV *News* (SR=10%) experiments. The first two rows: the visual compositions (31-th, 20-th, and 10-th spectral bands as R, G, and B channels, respectively) and the corresponding residual images for the MSI *Toy* data; The bottom two rows: the visual compositions (13-th frame) and the corresponding residual images for the CV *News* data.

Table 1: Quality metric (i.e., MPSNR) values and average computational costs for all compared methods under three SRs: 5%, 10%, 20%. (Bold: best; Underline: second best)

| Data Index | $-$ | Method | | | | | | |
|---|---|---|---|---|---|---|---|---|
| | | Observed | HaLRTC [18] | t-SVD [33] | TMacTT [3] | TRLRF [32] | FCTN-TC [37] | TW-TC |
| *Toy* | 5% | 11.156 | 19.446 | 25.379 | 27.596 | 30.126 | 29.446 | **30.689** |
| | 10% | 11.391 | 24.087 | 29.527 | 32.134 | 35.428 | 34.169 | **37.121** |
| | 20% | 11.904 | 29.944 | 35.097 | 36.791 | 41.010 | 40.453 | **44.009** |
| | Time (s) | – | 4.74 | 15.72 | 63.50 | 181.25 | 26.27 | 154.67 |
| *News* | 5% | 8.806 | 15.185 | 26.791 | 25.972 | 26.942 | 25.064 | **28.887** |
| | 10% | 9.041 | 19.579 | 28.748 | 29.213 | 29.305 | 30.272 | **32.551** |
| | 20% | 9.553 | 23.935 | 34.533 | 32.367 | 32.923 | 34.897 | **36.206** |
| | Time (s) | – | 7.64 | 31.76 | 36.40 | 521.30 | 128.62 | 342.36 |
| *Container* | 5% | 4.600 | 18.273 | 27.979 | 23.681 | 27.333 | 28.364 | **29.473** |
| | 10% | 4.834 | 21.198 | 31.690 | 26.646 | 28.834 | 33.721 | **34.459** |
| | 20% | 5.344 | 24.901 | 35.460 | 35.204 | 35.726 | 37.536 | **38.259** |
| | Time (s) | – | 8.88 | 23.70 | 71.35 | 204.78 | 121.31 | 339.04 |
| *HSV* | 5% | 7.494 | 11.579 | 39.736 | 42.459 | 42.501 | 42.675 | **48.999** |
| | 10% | 7.729 | 22.795 | 44.756 | 47.563 | 47.132 | 49.067 | **52.743** |
| | 20% | 8.240 | 32.631 | 50.123 | 51.065 | 51.961 | 53.566 | **54.678** |
| | Time (s) | – | 9.32 | 19.57 | 144.98 | 459.79 | 101.77 | 493.38 |

**Video Data.** The tested video data contains two color videos[3] (CVs): *News* and *Container*, and one hyperspectral video[4] (HSV) [20]. The former two are of size $144 \times 176 \times 3 \times 20$ (i.e., height $\times$ width $\times$ channel $\times$ frame), and the latter is of size $40 \times 40 \times 20 \times 20$ (i.e., height $\times$ width $\times$ spectral $\times$ frame). When tested on two CVs, we also empirically set $R_3 = R_4$ and $L_1 = L_2 = L_3 = L_4$, and then specify ranks $R_1$, $R_2$, $R_3$ and $L_1$ from the candidate sets $\{2, 3, 4, 5\}$, $\{10, 15, 20, 25\}$, $\{3, 4, 5\}$ and $\{2, 3\}$, respectively. More directly, we designate $R_1 = R_2 = R_3 = R_4 = 6$ and $L_1 = L_2 = L_3 = L_4 = 4$ to the HSV data experiment for all three SRs, aiming to alleviate the challenge of determining TW-ranks for these data with more balanced dimensions. The numerical and graphical results of all compared methods are exhibited in Table 1 and Figure 4, respectively. From the results, we detect that the TW-TC method invariably contributes the best performances and further enhanced advantages, which jointly illustrates the superiority of the TW topology for higher-order tensor characterization.

Besides, further studies, e.g., the effect investigations of TW-ranks, are conducted in Appendix D.

---

[3]http://trace.eas.asu.edu/yuv/
[4]https://openremotesensing.net/knowledgebase/

# 4 Conclusion

This paper suggested a novel TN decomposition, i.e., TW decomposition, and presented an ALS-based algorithm to learn such a decomposition iteratively. The TW decomposition verified theoretically and numerically potential for more accurately characterizing the complex interactions inside higher-order tensors using only linear hyper-parameter scaling, significantly contributing to the advancement of TNs in machine learning. Moreover, we provided a TW decomposition-based TC model, i.e., TW-TC, for high-order tensor data recovery and developed an efficient PAM-based solving algorithm, which enjoys a convergence guarantee. Substantial experiments on synthetic and real-world data confirmed that the TW-TC model markedly outperforms other decomposition-based state-of-the-art methods.

## Acknowledgments and Disclosure of Funding

The authors would like to thank the area chair and anonymous reviewers for their insightful comments and constructive suggestions. This research is supported by NSFC (Grant Nos. 12171072, 12271083), Natural Science Foundation of Sichuan Province (Grant No. 2022NSFSC0501), Key Projects of Applied Basic Research in Sichuan Province (Grant No. 2020YJ0216), National Key Research and Development Program of China (Grant No. 2020YFA0714001), Major Key Project of PCL (Grant No. PCL2021A12), and China NSFC projects under contract (Grant No. U1811461).

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
