# Supplementary Material of "Tensor Wheel Decomposition and Its Tensor Completion Application"

**Zhong-Cheng Wu**[1], **Ting-Zhu Huang**[*,1], **Liang-Jian Deng**[*,1], **Hong-Xia Dou**[2], **Deyu Meng**[3,4]

[1]School of Mathematical Sciences, University of Electronic Science and Technology of China
[2]School of Science, Xihua University
[3]School of Mathematics and Statistics, Xi'an Jiaotong University
[4]Pazhou Laboratory (Huangpu)

wuzhch97@163.com, tingzhuhuang@126.com, liangjian.deng@uestc.edu.cn
hongxia.dou@mail.xhu.edu.cn, dymeng@mail.xjtu.edu.cn

The supplementary material is organized as follows:

- Appendix A provides complete proofs of Theorems 1-4 and Lemmas 1-4.
- Appendix B presents more details for solving the proposed TW-TC model.
- Appendix C supplements all parameter configurations and some result analysis for the synthetic data experiments.
- Appendix D establishes additional discussion experiments, e.g., investigating the effects of TW-ranks.
- Appendix E releases the MATLAB code for TW-TC model.

## A  Proofs of Theorems and Lemmas

### A.1  Proofs of Theorems

**Theorem 1** (**Core-Centered Circular Invariance**). *Given an $N$th-order tensor $\mathcal{X} \in \mathbb{R}^{I_1 \times I_2 \times \cdots \times I_N}$ and its TW decomposition $TW[\![\{\mathcal{G}_k\}_{k=1}^N; \mathcal{C}]\!]$. Assume that $\boldsymbol{n} = (n_1, n_2, \cdots, n_N)$ is the circular reordering of vector $(1, 2, \cdots, N)$, then the core-centered invariance gives $\vec{\mathcal{X}}^{\boldsymbol{n}} = TW[\![\{\mathcal{G}_{n_k}\}_{k=1}^N; \vec{\mathcal{C}}^{\boldsymbol{n}}]\!]$.*

*Proof.* Since $TW[\![\{\mathcal{G}_k\}_{k=1}^N; \mathcal{C}]\!]$ is the TW decomposition of tensor $\mathcal{X} \in \mathbb{R}^{I_1 \times I_2 \times \cdots \times I_N}$, we have the $k$-contraction as follows,

$$\mathcal{X} = \mathcal{G}_1 \times_1^4 \cdots \times_1^{2k} \mathcal{G}_k \times_1^{2k+2} \cdots \times_{1,4}^{2N,1} \mathcal{G}_N \times_{1,2,\cdots,N}^{2,4,\cdots,2N} \mathcal{C}. \tag{1}$$

Similarly, when $\boldsymbol{n} = (n_1, n_2, \cdots, n_N)$ is obtained by circularly reordering vector $(1, 2, \cdots, N)$, we easily have

$$\vec{\mathcal{X}}^{\boldsymbol{n}} = \mathcal{G}_{n_1} \times_1^4 \cdots \times_1^{2k} \mathcal{G}_{n_k} \times_1^{2k+2} \cdots \times_{1,4}^{2N,1} \mathcal{G}_{n_N} \times_{1,2,\cdots,N}^{2,4,\cdots,2N} \vec{\mathcal{C}}^{\boldsymbol{n}}. \tag{2}$$

Thus, $\vec{\mathcal{X}}^{\boldsymbol{n}} = TW[\![\{\mathcal{G}_{n_k}\}_{k=1}^N; \vec{\mathcal{C}}^{\boldsymbol{n}}]\!]$ is clearly established. $\square$

**Theorem 2** (**Core-Connected Invariance**). *Assume that the TW decomposition of $\mathcal{X} \in \mathbb{R}^{I_1 \times \cdots \times I_N}$ is $\vec{\mathcal{X}}^{\boldsymbol{n}} = TW[\![\{\mathcal{G}_{n_k}\}_{k=1}^N; \vec{\mathcal{C}}^{\boldsymbol{n}}]\!]$, where $\boldsymbol{n} = (n_1, n_2, \cdots, n_N)$ is any vector that circularly shifts vector $(1, 2, \cdots, N)$. Let vector $\boldsymbol{e} = (n_1, n_k, n_2, \cdots, n_{k-1}, n_{k+1}, \cdots, n_N)$ $(3 \leq k < N, \; k \in \mathbb{Z})$, then*

$$\vec{\mathcal{X}}^{\boldsymbol{e}} = (\mathcal{G}_{n_1} \times_1^3 \vec{\mathcal{C}}^{\boldsymbol{e}} \times_3^4 \mathcal{G}_{n_k}) \times_{1,3,\cdots,2k-3,2k-2}^{3,4,\cdots,k+1,N+2} \mathcal{U}_{n_2,\cdots,n_{k-1}} \times_{2(N-k)+2,3,5,\cdots,2(N-k)+1,1}^{1,3,4,\cdots,N-k+2,N-k+4} \mathcal{V}_{n_{k+1},\cdots,n_N},$$

*where $\mathcal{U}_{n_2,\cdots,n_{k-1}} = \mathcal{G}_{n_2} \times_1^4 \cdots \times_1^{2(k-2)} \mathcal{G}_{n_{k-1}}$ and $\mathcal{V}_{n_{k+1},\cdots,n_N} = \mathcal{G}_{n_{k+1}} \times_1^4 \cdots \times_1^{2(N-k)} \mathcal{G}_{n_N}$.*

---

*Corresponding authors.

*Proof.* Since $\vec{\mathcal{X}}^{\boldsymbol{n}} = \mathrm{TW}[\![\{\mathcal{G}_{n_k}\}_{k=1}^N; \vec{\mathcal{C}}^{\boldsymbol{n}}]\!]$, we easily have

$$\mathcal{U}_{n_2,\cdots,n_{k-1}} = \mathcal{G}_{n_2} \times_1^4 \cdots \times_1^{2(k-2)} \mathcal{G}_{n_{k-1}} \in \mathbb{R}^{R_{n_2} \times I_{n_2} \times L_{n_2} \times \cdots \times I_{n_{k-1}} \times L_{n_{k-1}} \times R_{n_k}}, \qquad (3)$$

and

$$\mathcal{V}_{n_{k+1},\cdots,n_N} = \mathcal{G}_{n_{k+1}} \times_1^4 \cdots \times_1^{2(N-k)} \mathcal{G}_{n_N} \in \mathbb{R}^{R_{n_{k+1}} \times I_{n_{k+1}} \times L_{n_{k+1}} \times \cdots \times I_{n_N} \times L_{n_N} \times R_{n_1}}. \qquad (4)$$

According to vector $\boldsymbol{e} = (n_1, n_k, n_2, \cdots, n_{k-1}, n_{k+1}, \cdots, n_N)$ $(3 \leq k < N, \; k \in \mathbb{Z})$, then the corresponding tensor contractions yield

$$\mathcal{G}_{n_1} \times_1^3 \vec{\mathcal{C}}^{\boldsymbol{e}} \times_3^4 \mathcal{G}_{n_k} \in \mathbb{R}^{R_{n_1} \times I_{n_1} \times R_{n_2} \times L_{n_2} \times \cdots \times L_{n_{k-1}} \times L_{n_{k+1}} \times \cdots \times L_{n_N} \times R_{n_k} \times I_{n_k} \times R_{n_{k+1}}}. \qquad (5)$$

Obviously, there are $k$ common modes between $\mathcal{G}_{n_1} \times_1^3 \vec{\mathcal{C}}^{\boldsymbol{e}} \times_3^4 \mathcal{G}_{n_k}$ and $\mathcal{U}_{n_2,\cdots,n_{k-1}}$, i.e., $R_{n_2}, L_{n_2}, \cdots, L_{n_{k-1}}$, and $R_{n_k}$. Eliminating the $k$ modes, we can obtain

$$(\mathcal{G}_{n_1} \times_1^3 \vec{\mathcal{C}}^{\boldsymbol{e}} \times_3^4 \mathcal{G}_{n_k}) \times_{1,3,\cdots,2k-3,2k-2}^{3,4,\cdots,k+1,N+2} \mathcal{U}_{n_2,\cdots,n_{k-1}}$$
$$\in \mathbb{R}^{R_{n_1} \times I_{n_1} \times L_{n_{k+1}} \times \cdots \times L_{n_N} \times I_{n_k} \times R_{n_{k+1}} \times I_{n_2} \times \cdots \times I_{n_{k-1}}}. \qquad (6)$$

Similarly, there are $N - k + 2$ common modes between the above generated tensor $(\mathcal{G}_{n_1} \times_1^3 \vec{\mathcal{C}}^{\boldsymbol{e}} \times_3^4 \mathcal{G}_{n_k}) \times_{1,3,\cdots,2k-3,2k-2}^{3,4,\cdots,k+1,N+2} \mathcal{U}_{n_2,\cdots,n_{k-1}}$ and $\mathcal{V}_{n_{k+1},\cdots,n_N}$, i.e., $R_{n_1}, L_{n_{k+1}}, \cdots, L_{n_N}$, and $R_{n_{k+1}}$. Again, all contractions along these modes lead to

$$(\mathcal{G}_{n_1} \times_1^3 \vec{\mathcal{C}}^{\boldsymbol{e}} \times_3^4 \mathcal{G}_{n_k}) \times_{1,3,\cdots,2k-3,2k-2}^{3,4,\cdots,k+1,N+2} \mathcal{U}_{n_2,\cdots,n_{k-1}} \times_{2(N-k)+2,3,5,\cdots,2(N-k)+1,1}^{1,3,4,\cdots,N-k+2,N-k+4} \mathcal{V}_{n_{k+1},\cdots,n_N}$$
$$\in \mathbb{R}^{I_{n_1} \times I_{n_k} \times I_{n_2} \times \cdots \times I_{n_{k-1}} \times I_{n_{k+1}} \times \cdots \times I_{n_N}}, \qquad (7)$$

which is definitely equivalent to the $\boldsymbol{e}$-based permutation of $\mathcal{X}$, i.e., $\vec{\mathcal{X}}^{\boldsymbol{e}}$. The proof is completed. $\qquad\square$

**Theorem 3** (**Tensor Subwheel Equation**). *Assume that the TW decomposition of $\mathcal{X} \in \mathbb{R}^{I_1 \times I_2 \times \cdots \times I_N}$ is $\vec{\mathcal{X}}^{\boldsymbol{n}} = TW[\![\{\mathcal{G}_{n_k}\}_{k=1}^N; \vec{\mathcal{C}}^{\boldsymbol{n}}]\!]$, where $\boldsymbol{n} = (n_1, n_2, \cdots, n_N)$ is any vector that circularly shifts vector $(1, 2, \cdots, N)$. Let $\boldsymbol{m} = (N+1, N+2, 1, 2, \cdots, N)$ and $\boldsymbol{v} = (2, 4, \cdots, 2N, 1, 3, \cdots, 2N-1)$, then there inherently exists the following two tensor subwheel equations,*

$$\mathbf{X}_{<n_N>} = (\mathbf{G}_{n_N})_{(2)}(\mathbf{M}_{\neq n_N})_{[\boldsymbol{m};3]} \quad and \quad \vec{\mathbf{x}}_{[1:N,0]}^{\boldsymbol{n}} = \vec{\mathbf{c}}_{[1:N,0]}^{\boldsymbol{n}}(\mathbf{N}_{\neq \mathcal{C}})_{[\boldsymbol{v};N]}, \qquad (8)$$

*where $\mathcal{M}_{\neq n_N} \in \mathbb{R}^{R_{n_1} \times I_{n_1} \times \cdots \times I_{n_{N-1}} \times R_{n_N} \times L_{n_N}}$ is an $(N+2)$th-order subwheel tensor, which merges all TW factors but $\mathcal{G}_{n_N}$, i.e., $\mathcal{M}_{\neq n_N} = \mathcal{G}_{n_1} \times_1^4 \cdots \times_1^{2k} \mathcal{G}_{n_k} \times_1^{2k+2} \cdots \times_1^{2N-2} \mathcal{G}_{n_{N-1}} \times_{1,2,\cdots,N-1}^{3,5,\cdots,2N-1} \vec{\mathcal{C}}^{\boldsymbol{n}}$, and $\mathcal{N}_{\neq \mathcal{C}} \in \mathbb{R}^{I_{n_1} \times L_{n_1} \times \cdots \times I_{n_N} \times L_{n_N}}$ is another $2N$th-order subwheel tensor obtained by only merging $\{\mathcal{G}_{n_k}\}_{k=1}^N$, i.e., $\mathcal{N}_{\neq \mathcal{C}} = \mathcal{G}_{n_1} \times_1^4 \cdots \times_1^{2k} \mathcal{G}_{n_k} \times_1^{2k+2} \cdots \times_{1,4}^{2N,1} \mathcal{G}_{n_N}$.*

*Proof.* Since $\vec{\mathcal{X}}^{\boldsymbol{n}} = \mathrm{TW}[\![\{\mathcal{G}_{n_k}\}_{k=1}^N; \vec{\mathcal{C}}^{\boldsymbol{n}}]\!]$ is the TW decomposition of $\vec{\mathcal{X}}^{\boldsymbol{n}} \in \mathbb{R}^{I_{n_1} \times I_{n_2} \times \cdots \times I_{n_N}}$, we rewrite its form of tensor $k$-contraction as follows,

$$\vec{\mathcal{X}}^{\boldsymbol{n}} = \mathcal{G}_{n_N} \times_{N+1,N+2,1}^{1,3,4} \mathcal{M}_{\neq n_N} \quad and \quad \vec{\mathcal{X}}^{\boldsymbol{n}} = \vec{\mathcal{C}}^{\boldsymbol{n}} \times_{2,4,\cdots,2N}^{1,2,\cdots,N} \mathcal{N}_{\neq \mathcal{C}}, \qquad (9)$$

where

$$\mathcal{M}_{\neq n_N} = \mathcal{G}_{n_1} \times_1^4 \cdots \times_1^{2k} \mathcal{G}_{n_k} \times_1^{2k+2} \cdots \times_1^{2N-2} \mathcal{G}_{n_{N-1}} \times_{1,2,\cdots,N-1}^{3,5,\cdots,2N-1} \vec{\mathcal{C}}^{\boldsymbol{n}} \qquad (10)$$

and

$$\mathcal{N}_{\neq \mathcal{C}} = \mathcal{G}_{n_1} \times_1^4 \cdots \times_1^{2k} \mathcal{G}_{n_k} \times_1^{2k+2} \cdots \times_{1,4}^{2N,1} \mathcal{G}_{n_N}. \qquad (11)$$

Following Definition 3 (i.e., Generalized Tensor $k$-Contraction) of the main text, then formula (9) can be converted as corresponding matrix products

$$\mathbf{X}_{<n_N>} = (\mathbf{G}_{n_N})_{(2)}(\mathbf{M}_{\neq n_N})_{[\boldsymbol{m};3]} \quad and \quad \vec{\mathbf{x}}_{[1:N,0]}^{\boldsymbol{n}} = \vec{\mathbf{c}}_{[1:N,0]}^{\boldsymbol{n}}(\mathbf{N}_{\neq \mathcal{C}})_{[\boldsymbol{v};N]}, \qquad (12)$$

where $\boldsymbol{m} = (N+1, N+2, 1, 2, \cdots, N)$ and $\boldsymbol{v} = (2, 4, \cdots, 2N, 1, 3, \cdots, 2N-1)$ are also required. This completes the proof. $\qquad\square$

**Theorem 4.** *Assume that $\mathcal{X} = TW[\![\{\mathcal{G}_k\}_{k=1}^N; \mathcal{C}]\!]$ with $N$ ring factors $\mathcal{G}_k \in \mathbb{R}^{R_k \times I_k \times L_k \times R_{k+1}}$, then*

$$\mathtt{Rank}(\mathbf{X}_{(k)}) = \mathtt{Rank}(\mathbf{X}_{<k>}) \leq L_k \prod_{i=k}^{k+1} R_i, \quad k = 1, 2, \cdots, N. \qquad (13)$$

*Proof.* According to Theorem 3, we have the subwheel equations for all $k = 1, 2, \cdots, N$, as follows,

$$\mathbf{X}_{<k>} = (\mathbf{G}_k)_{(2)}(\mathbf{M}_{\neq k})_{[\boldsymbol{m};3]}. \tag{14}$$

Then, we can justify the following inequality,

$$
\begin{aligned}
\texttt{Rank}(\mathbf{X}_{(k)}) = \texttt{Rank}(\mathbf{X}_{<k>}) &\leq \min\{\texttt{Rank}((\mathbf{G}_k)_{(2)}), \texttt{Rank}((\mathbf{M}_{\neq k})_{[\boldsymbol{m};3]})\} \\
&\leq \texttt{Rank}((\mathbf{G}_k)_{(2)}) \\
&\leq \min\{I_k, R_k L_k R_{k+1}\} \\
&\leq L_k \prod_{i=k}^{k+1} R_i.
\end{aligned} \tag{15}
$$

The proof is completed. $\qquad\square$

## A.2 Proofs of Lemmas

Before proving the following Lemmas 1-4, we firstly rewrite the proposed TW-TC model as follows,

$$\min_{\mathcal{Z}} \ \Phi(\mathcal{Z}) = h(\mathcal{Z}) + f(\mathcal{X}), \tag{16}$$

where $\mathcal{Z} = (\mathcal{X}, \mathcal{G}_{1:N}, \mathcal{C})$, $h(\mathcal{Z}) = 1/2\|\mathcal{X} - TW[\![\{\mathcal{G}_k\}_{k=1}^N; \mathcal{C}]\!]\|_F^2$, and $f(\mathcal{X}) = \iota(\mathcal{X})$.

**Lemma 1.** *The objective function $\Phi(\mathcal{Z})$ in formula (16) is a Kurdyka-Łojasiewicz (KŁ) function.*

*Proof.* Following [1, 2], since $h(\mathcal{Z})$ is a polynomial function of $N + 2$ coupling variations, it is an obviously real-analytic function. Regarding the $f(\mathcal{X})$, since the constraint set onto $\{\mathcal{L} : \mathcal{P}_\Omega(\mathcal{L}) = \mathcal{P}_\Omega(\mathcal{F})\}$ is semi-algebraic, the $\iota(\mathcal{X})$ is a semi-algebraic function resulting from that indicator functions of semi-algebraic sets are semi-algebraic functions. Thus, $\Phi(\mathcal{Z})$, as a finite sum of real-valued analytic and semi-algebraic functions, is a KŁ function. $\qquad\square$

**Lemma 2 (Sufficient Decrease Condition).** *Let $\{\mathcal{Z}^{(t)}\}_{t \in \mathbb{N}}$ be the sequence generated by Algorithm 2 of the main text. Then, the sequence $\{\Phi(\mathcal{Z}^{(t)})\}_{t \in \mathbb{N}}$ explicitly satisfies the nonincreasing inequality,*

$$\Phi(\mathcal{Z}^{(t)}) - \Phi(\mathcal{Z}^{(t+1)}) \geq \frac{\rho}{2}\|\mathcal{Z}^{(t+1)} - \mathcal{Z}^{(t)}\|_F^2,$$

*where* $\|\mathcal{Z}^{(t+1)} - \mathcal{Z}^{(t)}\|_F^2 = \|\mathcal{X}^{(t+1)} - \mathcal{X}^{(t)}\|_F^2 + \sum_{k=1}^N \|\mathcal{G}_k^{(t+1)} - \mathcal{G}_k^{(t)}\|_F^2 + \|\mathcal{C}^{(t+1)} - \mathcal{C}^{(t)}\|_F^2.$

*Proof.* Since the sequences $\{\mathcal{G}_k^{(t)}\}_{t \in \mathbb{N}}$, $k = 1, 2, \cdots, N$, are generated by minimizing

$$\frac{1}{2}\|\mathcal{X}^{(t)} - TW[\![\mathcal{G}_{1:k-1}^{(t+1)}, \mathcal{G}_k, \mathcal{G}_{k+1:N}^{(t)}; \mathcal{C}^{(t)}]\!]\|_F^2 + \frac{\rho}{2}\|\mathcal{G}_k - \mathcal{G}_k^{(t)}\|_F^2, \quad k = 1, 2, \cdots, N, \tag{17}$$

each $\mathcal{G}_k^{(t+1)}$ is a minimizer of (17) in $(t+1)$-th iteration, thus leading to inequalities as follows,

$$h(\mathcal{X}^{(t)}, \mathcal{G}_{1:k}^{(t+1)}, \mathcal{G}_{k+1:N}^{(t)}, \mathcal{C}^{(t)}) + \frac{\rho}{2}\|\mathcal{G}_k^{(t+1)} - \mathcal{G}_k^{(t)}\|_F^2 \leq h(\mathcal{X}^{(t)}, \mathcal{G}_{1:k-1}^{(t+1)}, \mathcal{G}_{k:N}^{(t)}, \mathcal{C}^{(t)}), \ k = 1, 2, \cdots, N. \tag{18}$$

Likewisely, the sequences $\{\mathcal{C}^{(t)}\}_{t \in \mathbb{N}}$ and $\{\mathcal{X}^{(t)}\}_{t \in \mathbb{N}}$ are generated by respectively minimizing

$$\frac{1}{2}\|\mathcal{X}^{(t)} - TW[\![\mathcal{G}_{1:N}^{(t+1)}; \mathcal{C}]\!]\|_F^2 + \frac{\rho}{2}\|\mathcal{C} - \mathcal{C}^{(t)}\|_F^2 \tag{19}$$

and

$$\frac{1}{2}\|\mathcal{X} - TW[\![\mathcal{G}_{1:N}^{(t+1)}; \mathcal{C}^{(t+1)}]\!]\|_F^2 + \frac{\rho}{2}\|\mathcal{X} - \mathcal{X}^{(t)}\|_F^2 + \iota(\mathcal{X}), \tag{20}$$

thus $\mathcal{C}^{(t+1)}$ and $\mathcal{X}^{(t+1)}$ can respectively satisfy

$$h(\mathcal{X}^{(t)}, \mathcal{G}_{1:N}^{(t+1)}, \mathcal{C}^{(t+1)}) + \frac{\rho}{2}\|\mathcal{C}^{(t+1)} - \mathcal{C}^{(t)}\|_F^2 \leq h(\mathcal{X}^{(t)}, \mathcal{G}_{1:N}^{(t+1)}, \mathcal{C}^{(t)}) \tag{21}$$

and

$$h(\mathcal{X}^{(t+1)}, \mathcal{G}_{1:N}^{(t+1)}, \mathcal{C}^{(t+1)}) + f(\mathcal{X}^{(t+1)}) + \frac{\rho}{2}\|\mathcal{X}^{(t+1)} - \mathcal{X}^{(t)}\|_F^2 \leq h(\mathcal{X}^{(t)}, \mathcal{G}_{1:N}^{(t+1)}, \mathcal{C}^{(t+1)}) + f(\mathcal{X}^{(t)}). \tag{22}$$

Especially, if $t \in \{t : t > 200 \text{ and } \mathtt{mode}(t,s) \neq 0\}$, then the equal sign in (21) is exactly established owing to $\mathcal{C}^{(t+1)} = \mathcal{C}^{(t)}$. By eliminating the duplicates on the left and right, we can deduce

$$
\begin{aligned}
\Phi(\mathcal{Z}^{(t)}) - \Phi(\mathcal{Z}^{(t+1)}) &\geq \frac{\rho}{2}\|\mathcal{X}^{(t+1)} - \mathcal{X}^{(t)}\|_F^2 + \sum_{k=1}^{N} \frac{\rho}{2}\|\mathcal{G}_k^{(t+1)} - \mathcal{G}_k^{(t)}\|_F^2 + \frac{\rho}{2}\|\mathcal{C}^{(t+1)} - \mathcal{C}^{(t)}\|_F^2 \\
&\geq \frac{\rho}{2}(\|\mathcal{X}^{(t+1)} - \mathcal{X}^{(t)}\|_F^2 + \sum_{k=1}^{N} \|\mathcal{G}_k^{(t+1)} - \mathcal{G}_k^{(t)}\|_F^2 + \|\mathcal{C}^{(t+1)} - \mathcal{C}^{(t)}\|_F^2) \\
&= \frac{\rho}{2}\|\mathcal{Z}^{(t+1)} - \mathcal{Z}^{(t)}\|_F^2.
\end{aligned}
\tag{23}
$$

The proof is completed. $\qquad\square$

**Lemma 3** (**Relative Error Condition**). *Let $\{\mathcal{Z}^{(t)}\}_{t\in\mathbb{N}}$ be the sequence generated by Algorithm 2 of the main text. Then, there exists*

$$
\|\partial\Phi(\mathcal{Z}^{(t+1)})\|_F \leq \{L_\Phi + (N+2)\rho\}\|\mathcal{Z}^{(t+1)} - \mathcal{Z}^{(t)}\|_F,
$$

*where $L_\Phi$ sums the Lipschitz constants of $\{\partial_{\mathcal{G}_k}\Phi(\mathcal{Z})\}_{k=1}^{N}$ and $\partial_{\mathcal{C}}\Phi(\mathcal{Z})$, i.e., $L_\Phi = \sum_{k=1}^{N} L_{\mathcal{G}_k} + L_{\mathcal{C}}$.*

*Proof.* According to the proof of Lemma 2, i.e., $\mathcal{G}_k^{(t+1)}$, $k = 1, 2, \cdots, N$, $\mathcal{C}^{(t+1)}$ and $\mathcal{X}^{(t+1)}$ are the minimum solutions, thus we have $f(\mathcal{X}^{(t+1)}) \equiv 0$ for avoiding $\Phi(\mathcal{Z}) \to \infty$. Especially, when $t > 200$, the variable $\mathcal{C}$ tends to be relatively stable, i.e., $\|\mathcal{C}^{(t+1)} - \mathcal{C}^{(t)}\|_F$ is sufficiently small. Thus, we assume that when $t \in \{t : t > 200 \text{ and } \mathtt{mode}(t,s) \neq 0\}$, $\mathcal{C}^{(j)}$, $j = t+2, t+3, \cdots, t+s$, can approximately share the first-order optimal condition of $\mathcal{C}^{(t+1)}$. According to the fact that minimum solutions must satisfy the first-order optimal conditions, i.e., the sub-gradient equations of the objective function, then for all $t \in \mathbb{N}$, we always have

$$
\begin{cases}
0 \in \partial_{\mathcal{G}_k} h(\mathcal{X}^{(t)}, \mathcal{G}_{1:k}^{(t+1)}, \mathcal{G}_{k+1:N}^{(t)}, \mathcal{C}^{(t)}) + \rho(\mathcal{G}_k^{(t+1)} - \mathcal{G}_k^{(t)}), \ k = 1, 2, \cdots, N, \\
0 \in \partial_{\mathcal{C}} h(\mathcal{X}^{(t)}, \mathcal{G}_{1:N}^{(t+1)}, \mathcal{C}^{(t+1)}) + \rho(\mathcal{C}^{(t+1)} - \mathcal{C}^{(t)}), \\
0 \in \partial_{\mathcal{X}} h(\mathcal{X}^{(t+1)}, \mathcal{G}_{1:N}^{(t+1)}, \mathcal{C}^{(t+1)}) + \rho(\mathcal{X}^{(t+1)} - \mathcal{X}^{(t)}).
\end{cases}
\tag{24}
$$

Based on the sub-differentiability property, i.e.,

$$
\begin{aligned}
\partial\Phi(\mathcal{Z}^{(t+1)}) = \big(&\partial_{\mathcal{X}}\Phi(\mathcal{X}^{(t+1)}, \mathcal{G}_{1:N}^{(t+1)}, \mathcal{C}^{(t+1)}), \partial_{\mathcal{G}_1}\Phi(\mathcal{X}^{(t+1)}, \mathcal{G}_{1:N}^{(t+1)}, \mathcal{C}^{(t+1)}), \\
&\partial_{\mathcal{G}_2}\Phi(\mathcal{X}^{(t+1)}, \mathcal{G}_{1:N}^{(t+1)}, \mathcal{C}^{(t+1)}), \cdots, \partial_{\mathcal{G}_N}\Phi(\mathcal{X}^{(t+1)}, \mathcal{G}_{1:N}^{(t+1)}, \mathcal{C}^{(t+1)}), \\
&\qquad\qquad\qquad\qquad \partial_{\mathcal{C}}\Phi(\mathcal{X}^{(t+1)}, \mathcal{G}_{1:N}^{(t+1)}, \mathcal{C}^{(t+1)})\big),
\end{aligned}
\tag{25}
$$

where

$$
\begin{cases}
\partial_{\mathcal{X}}\Phi(\mathcal{X}^{(t+1)}, \mathcal{G}_{1:N}^{(t+1)}, \mathcal{C}^{(t+1)}) = \partial_{\mathcal{X}} h(\mathcal{X}^{(t+1)}, \mathcal{G}_{1:N}^{(t+1)}, \mathcal{C}^{(t+1)}), \\
\partial_{\mathcal{G}_k}\Phi(\mathcal{X}^{(t+1)}, \mathcal{G}_{1:N}^{(t+1)}, \mathcal{C}^{(t+1)}) = \partial_{\mathcal{G}_k} h(\mathcal{X}^{(t+1)}, \mathcal{G}_{1:N}^{(t+1)}, \mathcal{C}^{(t+1)}), \ k = 1, 2, \cdots, N, \\
\partial_{\mathcal{C}}\Phi(\mathcal{X}^{(t+1)}, \mathcal{G}_{1:N}^{(t+1)}, \mathcal{C}^{(t+1)}) = \partial_{\mathcal{C}} h(\mathcal{X}^{(t+1)}, \mathcal{G}_{1:N}^{(t+1)}, \mathcal{C}^{(t+1)}),
\end{cases}
\tag{26}
$$

then we have the triangle inequality as follows,

$$
\begin{aligned}
\|\partial\Phi(\mathcal{Z}^{(t+1)})\|_F \leq &\sum_{k=1}^{N} \|\partial_{\mathcal{G}_k} h(\mathcal{X}^{(t+1)}, \mathcal{G}_{1:N}^{(t+1)}, \mathcal{C}^{(t+1)})\|_F + \|\partial_{\mathcal{C}} h(\mathcal{X}^{(t+1)}, \mathcal{G}_{1:N}^{(t+1)}, \mathcal{C}^{(t+1)})\|_F \\
&+ \|\partial_{\mathcal{X}} h(\mathcal{X}^{(t+1)}, \mathcal{G}_{1:N}^{(t+1)}, \mathcal{C}^{(t+1)})\|_F.
\end{aligned}
\tag{27}
$$

Substitute into the first-order optimal condition (24), leading to

$$\|\partial\Phi(\mathcal{Z}^{(t+1)})\|_F$$

$$\leq \sum_{k=1}^{N} \|\partial_{\mathcal{G}_k}h(\mathcal{X}^{(t+1)},\mathcal{G}_{1:N}^{(t+1)},\mathcal{C}^{(t+1)}) - \partial_{\mathcal{G}_k}h(\mathcal{X}^{(t)},\mathcal{G}_{1:k}^{(t+1)},\mathcal{G}_{k+1:N}^{(t)},\mathcal{C}^{(t)}) - \rho(\mathcal{G}_k^{(t+1)} - \mathcal{G}_k^{(t)})\|_F$$

$$+ \|\partial_{\mathcal{C}}h(\mathcal{X}^{(t+1)},\mathcal{G}_{1:N}^{(t+1)},\mathcal{C}^{(t+1)}) - \partial_{\mathcal{C}}h(\mathcal{X}^{(t)},\mathcal{G}_{1:N}^{(t+1)},\mathcal{C}^{(t+1)}) - \rho(\mathcal{C}^{(t+1)} - \mathcal{C}^{(t)})\|_F$$

$$+ \rho\|\mathcal{X}^{(t+1)} - \mathcal{X}^{(t)}\|_F$$

$$\leq \sum_{k=1}^{N} \|\partial_{\mathcal{G}_k}h(\mathcal{X}^{(t+1)},\mathcal{G}_{1:N}^{(t+1)},\mathcal{C}^{(t+1)}) - \partial_{\mathcal{G}_k}h(\mathcal{X}^{(t)},\mathcal{G}_{1:k}^{(t+1)},\mathcal{G}_{k+1:N}^{(t)},\mathcal{C}^{(t)})\|_F$$

$$+ \sum_{k=1}^{N} \rho\|\mathcal{G}_k^{(t+1)} - \mathcal{G}_k^{(t)}\|_F + \|\partial_{\mathcal{C}}h(\mathcal{X}^{(t+1)},\mathcal{G}_{1:N}^{(t+1)},\mathcal{C}^{(t+1)}) - \partial_{\mathcal{C}}h(\mathcal{X}^{(t)},\mathcal{G}_{1:N}^{(t+1)},\mathcal{C}^{(t+1)})\|_F$$

$$+ \rho\|\mathcal{C}^{(t+1)} - \mathcal{C}^{(t)}\|_F + \rho\|\mathcal{X}^{(t+1)} - \mathcal{X}^{(t)}\|_F.$$

(28)

Since $h \in C^1$ and $f \equiv 0$, the partial derivatives $\{\partial_{\mathcal{G}_k}\Phi(\mathcal{Z})\}_{k=1}^N$ and $\partial_{\mathcal{C}}\Phi(\mathcal{Z})$ can be assumed to be $\{L_{\mathcal{G}_k}\}_{k=1}^N$ and $L_{\mathcal{C}}$-Lipschitz continuous, respectively. Then, we easily have

$$\begin{cases} \|\partial_{\mathcal{G}_k}h(\mathcal{X}^{(t+1)},\mathcal{G}_{1:N}^{(t+1)},\mathcal{C}^{(t+1)}) - \partial_{\mathcal{G}_k}h(\mathcal{X}^{(t)},\mathcal{G}_{1:k}^{(t+1)},\mathcal{G}_{k+1:N}^{(t)},\mathcal{C}^{(t)})\|_F \\ \quad \leq L_{\mathcal{G}_k}\|(\mathcal{X}^{(t+1)} - \mathcal{X}^{(t)}, \{\mathcal{G}_m^{(t+1)} - \mathcal{G}_m^{(t)}\}_{m=k+1}^N, \mathcal{C}^{(t+1)} - \mathcal{C}^{(t)})\|_F, \ k=1,2,\cdots,N, \quad (29) \\ \|\partial_{\mathcal{C}}h(\mathcal{X}^{(t+1)},\mathcal{G}_{1:N}^{(t+1)},\mathcal{C}^{(t+1)}) - \partial_{\mathcal{C}}h(\mathcal{X}^{(t)},\mathcal{G}_{1:N}^{(t+1)},\mathcal{C}^{(t+1)})\|_F \leq L_{\mathcal{C}}\|\mathcal{X}^{(t+1)} - \mathcal{X}^{(t)}\|_F. \end{cases}$$

Thus, a backsubstitution yields

$$\|\partial\Phi(\mathcal{Z}^{(t+1)})\|_F \leq \sum_{k=1}^{N} L_{\mathcal{G}_k}\|(\mathcal{X}^{(t+1)} - \mathcal{X}^{(t)}, \{\mathcal{G}_m^{(t+1)} - \mathcal{G}_m^{(t)}\}_{m=k+1}^N, \mathcal{C}^{(t+1)} - \mathcal{C}^{(t)})\|_F$$

$$+ \sum_{k=1}^{N} \rho\|(\mathcal{G}_k^{(t+1)} - \mathcal{G}_k^{(t)})\|_F + L_{\mathcal{C}}\|\mathcal{X}^{(t+1)} - \mathcal{X}^{(t)}\|_F + \rho\|\mathcal{C}^{(t+1)} - \mathcal{C}^{(t)}\|_F$$

$$+ \rho\|\mathcal{X}^{(t+1)} - \mathcal{X}^{(t)}\|_F$$

$$\leq \left(\sum_{k=1}^{N} L_{\mathcal{G}_k} + L_{\mathcal{C}}\right)\|\mathcal{Z}^{(t+1)} - \mathcal{Z}^{(t)}\|_F + (N+2)\rho\|\mathcal{Z}^{(t+1)} - \mathcal{Z}^{(t)}\|_F$$

$$= \{L_\Phi + (N+2)\rho\}\|\mathcal{Z}^{(t+1)} - \mathcal{Z}^{(t)}\|_F,$$

where $L_\Phi = \sum_{k=1}^N L_{\mathcal{G}_k} + L_{\mathcal{C}}$. The relative error condition is proved. $\qquad\square$

**Lemma 4.** *Let $\{\mathcal{Z}^{(t)}\}_{t\in\mathbb{N}}$ be the sequence generated by Algorithm 2 of the main text, then it is bounded.*

*Proof.* Relying upon the optimal solution in Algorithm 2 of the main text, i.e.,

$$\mathcal{X}^{(t+1)} = \mathcal{P}_{\Omega^c}\left(\frac{\text{TW}[\![\{\mathcal{G}_k^{(t+1)}\}_{k=1}^N; \mathcal{C}^{(t+1)}]\!] + \rho\mathcal{X}^{(t)}}{1+\rho}\right) + \mathcal{P}_\Omega(\mathcal{F}),$$

(30)

which requires that $\mathcal{X}$ is forcibly projected into set $\{\mathcal{L} : \mathcal{P}_\Omega(\mathcal{L}) = \mathcal{P}_\Omega(\mathcal{F})\}$. Thus, the indicator function $\iota(\mathcal{X}) \equiv 0$, i.e., $f(\mathcal{X}) \equiv 0$ in (16). As shown in Algorithm 2 of the main text, the initial $\mathcal{Z}^{(0)} = (\mathcal{X}^{(0)}, \mathcal{G}_{1:N}^{(0)}, \mathcal{C}^{(0)})$ is bounded, then $\Phi(\mathcal{Z}^{(0)})$ is bounded. According to Lemma 2, the sequence $\{\Phi(\mathcal{Z}^{(t)})\}_{t\in\mathbb{N}}$ decreases sufficiently, thus leading to $0 \leq \Phi(\mathcal{Z}^{(t)}) \leq \Phi(\mathcal{Z}^{(0)})$, i.e., $0 \leq h(\mathcal{Z}^{(t)}) \leq h(\mathcal{Z}^{(0)})$ for $\forall t \in \mathbb{N}$. Since the continuous function $h(\mathcal{Z})$ is proper and coercive, there exists $\|\mathcal{Z}\|_F \to \infty$ if and only if $h(\mathcal{Z}) \to \infty$. Obviously, the discrete points $h(\mathcal{Z}^{(t)}) \not\to \infty$, thus $\|\mathcal{Z}^{(t)}\|_F \not\to \infty$, i.e., sequence $\{\mathcal{Z}^{(t)}\}_{t\in\mathbb{N}}$ is certainly bounded. $\qquad\square$

Table 1: The parameter configurations of all compared methods for the synthetic data experiments under different cases.

| Data Type | SR | Method | Rank Setting | Storage Complexity $\mathcal{O}(\text{Ave.} \pm \text{Std.})$ |
|---|---|---|---|---|
| *Third-order tensor* | 20% | TT-TC (PAM) | $(23, 8)$ | $\mathcal{O}(11288 \pm 1046)$ |
| | | TR-TC (PAM) | $(4, 6, 5)$ | $\mathcal{O}(3885 \pm 244)$ |
| | | FCTN-TC (PAM) | $(4, 5, 5)$ | $\mathcal{O}(3413 \pm 213)$ |
| | | Tucker-TC (PAM) | $(6, 6, 6)$ | $\mathcal{O}(1161 \pm 59)$ |
| | | TW-TC (PAM) | $(2, 3, 4, 2, 2, 3)$ | $\mathcal{O}(3162 \pm 203)$ |
| | 40% | TT-TC (PAM) | $(26, 14)$ | $\mathcal{O}(21210 \pm 2058)$ |
| | | TR-TC (PAM) | $(10, 2, 8)$ | $\mathcal{O}(6090 \pm 473)$ |
| | | FCTN-TC (PAM) | $(8, 8, 2)$ | $\mathcal{O}(5040 \pm 382)$ |
| | | Tucker-TC (PAM) | $(6, 6, 6)$ | $\mathcal{O}(1161 \pm 59)$ |
| | | TW-TC (PAM) | $(7, 2, 2, 2, 2, 2)$ | $\mathcal{O}(3368 \pm 228)$ |
| *Fourth-order tensor* | 20% | TT-TC (PAM) | $(17, 11, 12)$ | $\mathcal{O}(6960 \pm 378)$ |
| | | TR-TC (PAM) | $(13, 9, 5, 5)$ | $\mathcal{O}(5040 \pm 236)$ |
| | | FCTN-TC (PAM) | $(3, 2, 5, 3, 2, 2)$ | $\mathcal{O}(1600 \pm 69)$ |
| | | Tucker-TC (PAM) | $(5, 5, 5, 5)$ | $\mathcal{O}(1025 \pm 16)$ |
| | | TW-TC (PAM) | $(4, 6, 3, 2, 2, 2, 2, 2)$ | $\mathcal{O}(2256 \pm 104)$ |
| | 40% | TT-TC (PAM) | $(20, 17, 6)$ | $\mathcal{O}(9360 \pm 584)$ |
| | | TR-TC (PAM) | $(19, 6, 2, 13)$ | $\mathcal{O}(7980 \pm 449)$ |
| | | FCTN-TC (PAM) | $(2, 5, 5, 4, 3, 3)$ | $\mathcal{O}(3580 \pm 153)$ |
| | | Tucker-TC (PAM) | $(5, 5, 5, 5)$ | $\mathcal{O}(1025 \pm 16)$ |
| | | TW-TC (PAM) | $(8, 7, 4, 3, 2, 2, 2, 2)$ | $\mathcal{O}(4816 \pm 224)$ |
| *Fifth-order tensor* | 20% | TT-TC (PAM) | $(11, 17, 6, 5)$ | $\mathcal{O}(2513 \pm 109)$ |
| | | TR-TC (PAM) | $(4, 21, 4, 5, 4)$ | $\mathcal{O}(1680 \pm 63)$ |
| | | FCTN-TC (PAM) | $(2, 2, 3, 3, 2, 2, 5, 3, 2, 2)$ | $\mathcal{O}(1470 \pm 46)$ |
| | | Tucker-TC (PAM) | $(3, 3, 3, 3, 3)$ | $\mathcal{O}(356 \pm 3)$ |
| | | TW-TC (PAM) | $(2, 2, 3, 2, 2, 2, 2, 2, 3, 2)$ | $\mathcal{O}(438 \pm 12)$ |
| | 40% | TT-TC (PAM) | $(23, 17, 7, 3)$ | $\mathcal{O}(4178 \pm 208)$ |
| | | TR-TC (PAM) | $(19, 15, 3, 2, 5)$ | $\mathcal{O}(3308 \pm 154)$ |
| | | FCTN-TC (PAM) | $(2, 3, 2, 2, 2, 2, 6, 3, 2, 2)$ | $\mathcal{O}(1350 \pm 43)$ |
| | | Tucker-TC (PAM) | $(3, 3, 3, 3, 3)$ | $\mathcal{O}(356 \pm 3)$ |
| | | TW-TC (PAM) | $(3, 5, 6, 3, 2, 2, 2, 2, 3, 2)$ | $\mathcal{O}(1218 \pm 40)$ |

# B  Solving Procedure for TW-TC Model

Under PAM framework [1], the proposed TW-TC model can be solved by

$$
\begin{cases}
\mathcal{G}_k^{(t+1)} \in \arg\min_{\mathcal{G}_k} \left\{ \frac{1}{2}\|\mathcal{X}^{(t)} - \text{TW}[\![\mathcal{G}_{1:k-1}^{(t+1)}, \mathcal{G}_k, \mathcal{G}_{k+1:N}^{(t)}; \mathcal{C}^{(t)}]\!]\|_F^2 + \frac{\rho}{2}\|\mathcal{G}_k - \mathcal{G}_k^{(t)}\|_F^2 \right\}, \\
\qquad\qquad\qquad\qquad\qquad\qquad\qquad\qquad\qquad\qquad\qquad\qquad k = 1, 2, \cdots, N, \\
\mathcal{C}^{(t+1)} \in \arg\min_{\mathcal{C}} \left\{ \frac{1}{2}\|\mathcal{X}^{(t)} - \text{TW}[\![\mathcal{G}_{1:N}^{(t+1)}; \mathcal{C}]\!]\|_F^2 + \frac{\rho}{2}\|\mathcal{C} - \mathcal{C}^{(t)}\|_F^2 \right\}, \\
\mathcal{X}^{(t+1)} \in \arg\min_{\mathcal{X}} \left\{ \frac{1}{2}\|\mathcal{X} - \text{TW}[\![\mathcal{G}_{1:N}^{(t+1)}; \mathcal{C}^{(t+1)}]\!]\|_F^2 + \frac{\rho}{2}\|\mathcal{X} - \mathcal{X}^{(t)}\|_F^2 + \iota(\mathcal{X}) \right\}.
\end{cases}
\tag{31}
$$

More specifically,

**Updating** $\mathcal{G}_k$, $k = 1, 2, \cdots, N$: Following Theorem 3, the matrix form of $\mathcal{G}_k$-subproblem can be given by

$$
(\mathbf{G}_k^{(t+1)})_{(2)} \in \arg\min_{(\mathbf{G}_k)_{(2)}} \left\{ \frac{1}{2}\|\mathbf{X}_{<k>}^{(t)} - (\mathbf{G}_k)_{(2)}(\mathbf{M}_{\neq k}^{(t)})_{[\boldsymbol{m};3]}\|_F^2 + \frac{\rho}{2}\|(\mathbf{G}_k)_{(2)} - (\mathbf{G}_k^{(t)})_{(2)}\|_F^2 \right\}, \tag{32}
$$

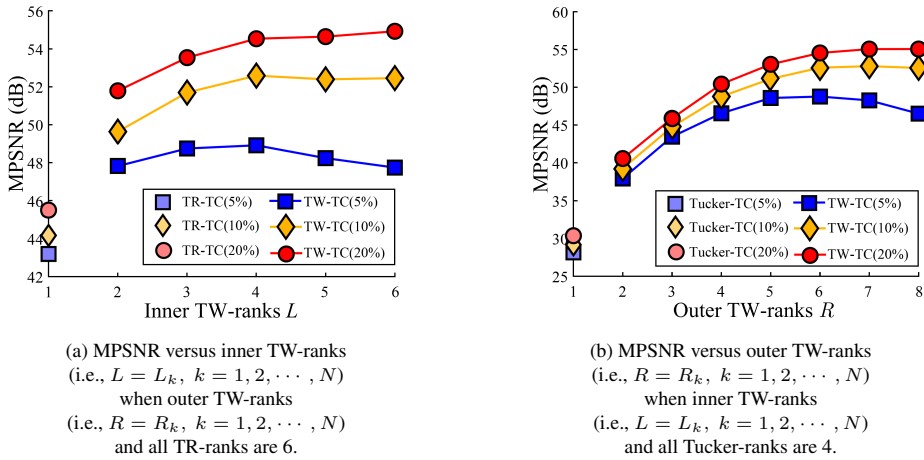

(a) MPSNR versus inner TW-ranks
(i.e., $L = L_k$, $k = 1, 2, \cdots, N$)
when outer TW-ranks
(i.e., $R = R_k$, $k = 1, 2, \cdots, N$)
and all TR-ranks are 6.

(b) MPSNR versus outer TW-ranks
(i.e., $R = R_k$, $k = 1, 2, \cdots, N$)
when inner TW-ranks
(i.e., $L = L_k$, $k = 1, 2, \cdots, N$)
and all Tucker-ranks are 4.

Figure 1: Recovery results of TR-TC, Tucker-TC, and TW-TC algorithms on HSV data under three SRs: 5%, 10%, 20%. Especially, the MPSNR values of TW-TC using various TW-ranks are recorded.

where $\mathcal{M}_{\neq k}^{(t)}$ and $\boldsymbol{m}$ are obtained relying upon Theorem 3. Thus, we easily have

$$\mathcal{G}_k^{(t+1)} = \texttt{Fold}_{(2)}\big\{(\mathbf{G}_k^{(t+1)})_{(2)}\big\} \tag{33}$$

with

$$(\mathbf{G}_k^{(t+1)})_{(2)} = \big(\mathbf{X}_{<k>}^{(t)}(\mathbf{M}_{\neq k}^{(t)})_{[\boldsymbol{m};3]}^T + \rho(\mathbf{G}_k^{(t)})_{(2)}\big)\big((\mathbf{M}_{\neq k}^{(t)})_{[\boldsymbol{m};3]}(\mathbf{M}_{\neq k}^{(t)})_{[\boldsymbol{m};3]}^T + \rho\mathbf{I}\big)^{-1}. \tag{34}$$

**Updating $\mathcal{C}$:** Similarly, the $\mathcal{C}$-subproblem is rewritten as

$$\mathbf{c}_{[1:N,0]}^{(t+1)} \in \arg\min_{\mathbf{c}_{[1:N,0]}} \Big\{ \frac{1}{2}\|\mathbf{x}_{[1:N,0]}^{(t)} - \mathbf{c}_{[1:N,0]}(\mathbf{N}_{\neq \mathcal{C}}^{(t)})_{[\boldsymbol{v};N]}\|_F^2 + \frac{\rho}{2}\|\mathbf{c}_{[1:N,0]} - \mathbf{c}_{[1:N,0]}^{(t)}\|_F^2 \Big\}, \tag{35}$$

where $\mathcal{N}_{\neq \mathcal{C}}^{(t)}$ and $\boldsymbol{v}$ are determined by Theorem 3. The above problem has a minimum solution, appearing as

$$\mathcal{C}^{(t+1)} = \texttt{Fold}_{[1:N,0]}\big\{\mathbf{c}_{[1:N,0]}^{(t+1)}\big\} \tag{36}$$

with

$$\mathbf{c}_{[1:N,0]}^{(t+1)} = \big(\mathbf{x}_{[1:N,0]}^{(t)}(\mathbf{N}_{\neq \mathcal{C}}^{(t)})_{[\boldsymbol{v};N]}^T + \rho\mathbf{c}_{[1:N,0]}^{(t)}\big)\big((\mathbf{N}_{\neq \mathcal{C}}^{(t)})_{[\boldsymbol{v};N]}(\mathbf{N}_{\neq \mathcal{C}}^{(t)})_{[\boldsymbol{v};N]}^T + \rho\mathbf{I}\big)^{-1}. \tag{37}$$

**Updating $\mathcal{X}$:** The $\mathcal{X}$-subproblem reduces to a least squares problem, thus its closed-form solution satisfies

$$\mathcal{X}^{(t+1)} = \mathcal{P}_{\Omega^c}\Big(\frac{\texttt{TW}[\![\{\mathcal{G}_k^{(t+1)}\}_{k=1}^N; \mathcal{C}^{(t+1)}]\!] + \rho\mathcal{X}^{(t)}}{1 + \rho}\Big) + \mathcal{P}_{\Omega}(\mathcal{F}), \tag{38}$$

where $\Omega^c$ indicates the complementary set of $\Omega$.

Although formulas (32) and (35) appear to be extremely complex, they are actually straightforward matrix least-squares issues. Thus, formulas (34) and (37) can be directly obtained by forcing the derivatives of formulas (32) and (35) w.r.t. $(\mathbf{G}_k)_{(2)}$ and $\mathbf{c}_{[1:N,0]}$ to 0, respectively. On another hand, the least-squares problem

$$\frac{1}{2}\|\mathcal{X} - \texttt{TW}[\![\mathcal{G}_{1:N}^{(t+1)}; \mathcal{C}^{(t+1)}]\!]\|_F^2 + \frac{\rho}{2}\|\mathcal{X} - \mathcal{X}^{(t)}\|_F^2 \tag{39}$$

has a closed-form solution as follows,

$$\mathcal{X} = \frac{\texttt{TW}[\![\{\mathcal{G}_k^{(t+1)}\}_{k=1}^N; \mathcal{C}^{(t+1)}]\!] + \rho\mathcal{X}^{(t)}}{1 + \rho}. \tag{40}$$

To make $\iota(\mathcal{X}) \equiv 0$, formula (40) is converted to formula (38).

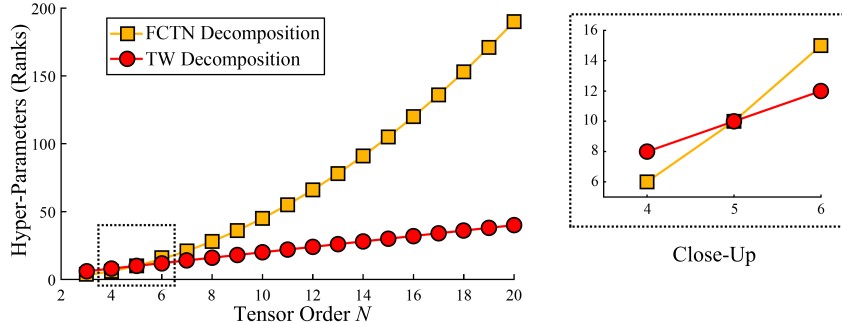

Figure 2: The number of hyper-parameters (i.e., ranks) of FCTN and TW decompositions against tensor dimension.

## C  Details for Experiments with Synthetic Data

### C.1  Parameter Details

To clearly understand the model complexity of all algorithms, Table 1 provides the parameter configurations and storage complexities. For each case, the hyper-parameters, i.e., TT-ranks, TR-ranks, FCTN-ranks, Tucker-ranks, and TW-ranks, are determined by respectively rounding the average of three corresponding rank collections, which are fine-tuned with a step size of 1 on three randomly selected experiments. Furthermore, the parameter storage complexity is also rounded.

### C.2  Result Analysis

The experimental descriptions are exhibited in the main text. We here explain some of the unintuitive results, i.e., the proposed TW-TC model requires more computing costs than the FCTN-TC one, and the TT-TC and TR-TC models perform better than the FCTN-TC one despite their fewer connections.

As empirically analyzed in [3], i.e., the required FCTN rank values are typically significantly lower than Tucker rank values. For the synthetic data experiments, these relatively small rank configurations experimentally yield the optimal performance of the FCTN-TC model (see Table 1) even if the obtained results are unsatisfactory (see Figure 3 of the main text), thus allowing a computational advantage over the TW-TC, TT-TC, and TR-TC ones. Moreover, as depicted in Figure 2, the proposed TW topology has a more complicated structure than the FCTN one under two exceptional cases, i.e., when the operated tensors are third-order or fourth-order. Consequently, the proposed TW-TC model requires incremental computational time in the synthetic data experiments, despite its higher recovery accuracy.

According to the optimal rank parameter configuration of the FCTN model, we argue that the inferior performance of FCTN-TC model may be caused by over-fitting, judged by its several optimal ranks between non-adjacent factors being 2 (see Table 1). Compared with real-world data, the low-rank characteristics of synthetic data are simpler. That is, not all non-adjacent dimensions have a direct relationship, thus leading the authentic ranks between some non-adjacent factors to be 1 (i.e., without connection). However, the minimum values of FCTN-ranks among several non-adjacent factors are 2 rather than 1, forcing the FCTN topology to maintain its fully-connected structure. Accordingly, some ineffective structures may reduce the performance of the FCTN-TC method, when applied to the synthesized fourth-order and fifth-order data.

## D  Discussion Experiments

In Section 2.3 of the main text, we briefly illustrate that TW decomposition can degenerate into TR and Tucker decompositions by enforcing the corresponding TW-ranks to be 1, showing its graphical potentiality over the latter. Numerically, we further confirm it by investigating the effects of TW-ranks.

Figure 1 provides the recovery results versus TW-ranks for the TR-TC (i.e., TR decomposition-based TC model), Tucker-TC (i.e., Tucker decomposition-based TC model), and the proposed TW-TC when SRs are 5%, 10%, 20%. These methods are uniformly solved by the PAM algorithm, and the HSV

data is adopted for presentation. At any considered SR, Figure 1(a) and Figure 1(b) experimentally indicate that TW decomposition compares favorably to TR and Tucker decompositions, respectively. Actually, the designed experiments also investigate how TW-ranks affect the quality of the TC results or, more internally, the characterization capability of TW decomposition. From Figure 1(a) and Figure 1(b), we observe that both inner TW-ranks $L$ and outer TW-ranks $R$ contribute positively to the final effect (i.e., MPSNR) within a certain range, implying the rationality of the TW topology without meaningless structure.

Moreover, as one of the motivations for designing the TW topology, Figure 2 shows the number of hyper-parameters of FCTN and TW decompositions (i.e., FCTN-ranks and TW-ranks, respectively) under different tensor dimensions. From Figure 2, we observe that our TW decomposition effectively alleviates the curse from dimensionality, thereby significantly improving its flexibility for higher-order tensor applications.

## E MATLAB Code

The code is available at: *https://github.com/zhongchengwu/code_TWDec*.