# OpenReview forum: "Tensor Wheel Decomposition and Its Tensor Completion Application"
_NeurIPS.cc/2022/Conference — NeurIPS 2022 Accept_

### Official Review · Reviewer_Dt2K · 2022-06-22

**Rating:** 5
**Confidence:** 4
**Soundness:** 2 fair
**Presentation:** 3 good
**Contribution:** 3 good

**Summary:**

In this paper, a new Tensor Network (TN) topology is proposed, the Tensor Wheel (TW), which can be seen as a combination of Tensor Ring (TR) and Tucker network topologies. The paper explores the properties of TW and develop a Proximal Alternating Minimization (PAM) algorithm to make inference of missing values in tensor data. Experimental results of tensor completion on synthetically generated and real-world signals are presented and compared against some state-of-the-art tensor completion methods.

**Questions:**

-	Could you please provide the details of the ranks used for each model used in Fig. 1 and Fig. 2?
-      Can you compute the number of model parameters in each case (sum of number of entries in each core tensor for each model)?
------------------------------------------------------
I updated my scores after rebuttal. See my comments below

**Limitations:**

Yes, the authors have stated that the main limitation of their proposed model is its exponentionally grow of model parameters with the number of dimensions.

**Strengths And Weaknesses:**

Strengths:
-	Clear definition of the new topology and description of some of its properties
Weaknesses
-	(major) I don’t agree with the limitation (ii) of current TN models: “At least one Nth-order factor is required to physically inherit the complex interactions from an Nth-order tensor”. TT and TR can model complex modes interactions if the ranks are large enough. The fact that there is a lack of direct connections from any pair of nodes is not a limitation because any nodes are fully connected through a TR or TT. However, the price to pay with TT or TR to model complex modes interactions is having bigger core tensor (larger number of parameters). The new proposed topology has also a large price to pay in terms of model size because the core tensor C grows exponentially with the number of dimensions, which makes it intractable in practice. The paper lacks from a comparison of TR/TT and TW for a fixed size of both models (see my criticism to experiments below).
-	The new proposed model can be used only with a small number of dimensions because of the curse of dimensionality imposed by the core tensor C.
-	(major) I think the proposed TW model is equivalent to TR by noting that, if the core tensor C is represented by a TR (this can be done always), then by fusing this TR with the cores G_n we can reach to TR representation equivalent to the former TW model. I would have liked to see this analysis in the paper and a discussion justifying TW over TR.
-	(major) Comparison against other models in the experiments are unclear. The value of the used ranks for all the models are omitted which make not possible a fair comparison. To show the superiority of TW over TT and TR, the authors must compare the tensor completion results for all the models but having the same number of model parameters. The number of model parameters can be computed by adding the number of entries of all core tensors for each model (see my question about experiment settings below).
-	(minor) The title should include the term “tensor completion” because that is the only application of the new model that is presented in the paper.
-	(minor) The absolute value operation in the definition of the Frobenius norm in line 77 is not needed because tensor entries are real numbers.
-	(minor) I don’t agree with the statement in line 163: “Apparently, the O(NIR^3+R^N) scales exponentially”. The exponential grow is not apparent, it is a fact.
------------------------------------------------------
I updated my scores after rebuttal. See my comments below

---

> ### Author Response · Authors · 2022-08-02
> **Rebuttal to all Comments.**
>
> Weakness 1: Dissenting descriptions regarding TR limitations.
>
> Response: In terms of tensor computation or representation, your comment is correct, i.e., more storage parameters will yield better decomposition performance. Thus, in tensor efficient computation, the parameter scale is not negligible when evaluating a new decomposition. However, we emphasize that the involved descriptions regarding TR limitations are mainly concerned with the characterization capabilities, which are crucial for TN models in high-order data recovery (line 60, page 2 of the revised manuscript). For high-order data recovery tasks, e.g., TC and denoising, the tensor decompositions are utilized as various predictors or fitters, whose topologies uniquely determine their characterization or fitting capabilities. Instead of aimlessly accumulating the number of parameters, the best results in the TC experiments are obtained by employing the optimal number of parameters. To avoid ambiguity, we have modified this mentioned sentence, see line 56, page 2 of the revised manuscript.
>
> Weakness 2: Lack of a comparison between TR/TT and TW for a fixed size of both models.
>
> Response: We consider that the experiment using a fixed size of both models may be superfluous. For the TC experiment, the best performance is usually achieved only with the proper rank parameters (i.e., an appropriate number of parameters). According to the revised Fig. 3 and Table 1 of the revised supplementary material, the proposed TW-TC model outperformed both the TT-TC and TR-TC models, while requiring fewer parameters. Therefore, we argue that such a suggested experiment may be redundant. Thanks for your valuable suggestion.
>
> Weakness 3: The proposed TW model is equivalent to TR.
>
> Response: We will explain this question from the following two folds. Firstly, after the tensor topology is determined, any re-decomposition or contraction will change the structural properties of the TN model. That is, such a changed decom- position is no longer the original one, they are not equivalent. Secondly, even if the core factor C is decomposed into TR factors, TW decomposition cannot be transformed into a TR decomposition after performing the corresponding contraction. This is because the newly obtained tensor topology has two layers of connections on adjacent fifth-order factors, which are different from those of TR decomposition. Consequently, we may disagree with this point.  Thanks for your feedback.
>
> Weakness 4: Comparisons to other models are unclear. The models' rank parameters are omitted.
>
> Response: Thanks for your constructive comment. Follwing your suggest, we have supplemented parameter configuration for all models in the synthetic data experiments (see Table 1 of the revised supplementary material), aiming to more clearly compare the model complexity from different algorithms. Also, we emphasize and discuss the comparison performance of all methods (see lines 249-256, page 8 in the revised manuscript).
>
> Weakness 5: The title should include the term “tensor completion”. The absolute value operation is not needed. Also, an inaccurate setence.
>
> Response: Thank you for helping improve our manuscript. For a more rigorous presentation, we have added the term "tensor completion" into the title, and removed the absolute value operation (see line 78, page 3 of the revised manuscript). Moreover, we have modified the ambiguous word "Apparently" as "It is clear that", leading to a new sentence (see line 164, page 5 of the revised manuscript).
>
> Question 1: Could you please provide the details of the ranks used for each model used in Fig. 1 and Fig. 2?
>
> Response: Since Fig. 1 and Fig. 2 are just two illustration figures that are not related to model ranks, perhaps you meant to ask about Figs. 3 and 4? If so, see the subsequent responses. As answered to Reviewer Bfwm, we cannot list all the rank parameters of different algorithms in Fig. 4, because some of them are non-decompositional, e.g., the HaLRTC [R5] (Tucker decomposition-based convex relaxation method), TMac-TT [R6] (TT decomposition-based parallel matrix decomposition method). However, we have wholly provided the details of the ranks for all models in Fig. 3 (see Table 1 of the revised supplementary material). Thanks for your this good question.
>
> Question 2: Can you compute the number of model parameters in each case?
>
> Response: We appreciate your question. Similar to Question 1, some of the methods used in Fig.4, e.g., the HaLRTC [R5], do not have the core factor tensor. Thus, we didn't compute the number of their model parameters under this case. For the models in Fig. 3, we have provided the number of model parameters for all models (see Table 1 of the revised supplementary material).
>
> [R5] Tensor completion for estimating missing values in visual data, IEEE TPAMI, 2013.
>
> [R6] Efficient tensor completion for color image and video recovery: Low-rank tensor train, IEEE TIP, 2016.

---

> > ### Comment · Reviewer_Dt2K · 2022-08-03
> > **I raised my scores based on new revision and response to my comments**
> >
> > Dear authors,
> > I really appreciate your efforts improving the paper and providing answers to all my concerns. Se below my comments to your responses.
> >
> > Weaknesss 1: Thanks for providing detailed responses that helped me to better understand your point of view.
> >
> > Weakness 2: I agree that for TC and denoising the highest compression ratio is not the main goal although intuitively higher compression ratio usually comes with higher predictability of missing/corrupted entries. Anyway, I accept your argument that in TC it is not necessary to compare different algorithms having the same size model. However, there are some issues with the experiments on synthetic data:
> > -	Synthetic data in Fig. 3 was generated using a Tucker model so it is expected that the algorithm based on the Tucker model outperform any other assumed model in terms of obtained error. Why Tucker algorithm performs so badly? This could be explained by some problem in the implementation of the Tucker algorithm.
> > -	Current Fig. 3 is not very helpful to my understanding because each algorithm should be tested on a dataset generated with the same model and, for example, some additive noise.
> >
> > Weakness. 3:  Sorry, I must disagree with your response or maybe there was a misunderstanding.  Let me explain a how a mathematical proof can be written to prove my affirmation. By decomposing core C into a TR (this is always possible), we obtain two connected concentric rings.  The inner ring has 3rd-order core tensors, while the outer ring has 4th-order core tensors. As a final step, we can contract the nodes connecting both rings which results in a single Tensor Ring topology.  Of course, the obtained TR has larger ranks than the ring in the original TW. I am not telling that TW is not an acceptable topology for inferring missing data in tensors, in fact, I think it is a valid proposal. I only wanted to highlight that TW can be seen as a TR topology, which is an interesting property that is missing in the current version of the paper.
> >
> > Weakness 4: Thank you for adding the parameter configuration for all models in the supplementary material. I think it is important for reproducibility and a more complete evaluation of the presented results.
> >
> > Weakness 5: Thanks for considering my suggestion of adding tensor completion in the title.

---

> > > ### Author Response · Authors · 2022-08-04
> > > **Rebuttal to the further Comments.**
> > >
> > > Thanks for your promote response and the positive comments to our rebuttal. We further respond to your two comments, as follows:
> > >
> > > Comment 1 (i.e., Weakness 2): Why Tucker algorithm performs so badly on the synthetic data generated using a Tucker model? Moreover, each algorithm should be tested on a dataset generated with the same model and, for example, some additive noise.
> > >
> > > Response: Thanks for your question. Mainly, the reason consists in two aspects: 1) the normalization of the tensor data (i.e., “All synthetic data are numerically renormalized into [0, 1]”), and 2) the local rather than global convergence of the Tucker-TC (PAM) algorithm. On Tucker factors-generated unprocessed data, the Tucker-TC (PAM) algorithm definitely can achieve the best recovery error ($10^{-16}$ to $10^{-10}$) compared to other algorithms, e.g., the suboptimal TW-TC (PAM) ($10^{-14}$ to $10^{-8}$). Nonetheless, such an evaluation may be worthless for the Tucker-TC (PAM) algorithm, since the experimental data unduly privileges it, which is usually impractical. Therefore, we renormalized all synthetic data into [0, 1] for a fairer comparison. Although these scaled data can still be accurately characterized by performing appropriate scaling on the Tucker factors, the local convergence of the Tucker-TC (PAM) algorithm only provides local rather than global minimizers owing to an immutable initialization (i.e., the uniform distribution U(0, 1)). After normalization, all the compared models are executed with the same initialization for fair comparison, while the proposed TW-TC model achieves the best performance.
> > >
> > > We regret that our experiments in Fig. 3 did not provide you with significant benefits. Actually, the suggested experiment design has been considered before. However, each model achieves remarkable results on the corresponding dataset generated by the same model, as in the case of Tucker, leading to obstacles in comparison. Moreover, since the experimental data for each model is different, we may not be able to explain whether better RES values indicate better performance. To the best of our knowledge, the synthetic data experiments in many excellent works, e.g., [R1, R2, R3, R4], are performed by comparing different methods on the same noise-free data, yielding an intuitive and comparable numerical result. Consequently, our experiments have a common setting with them and may be more suitable for this manuscript. Again, I appreciate your insightful comment.
> > >
> > > Comment 2 (i.e., Weakness 3): Insisting on an interesting property, TW can be seen as a TR topology.
> > >
> > > Response: Thanks for your efforts for such a detailed explanation. After careful understanding and thorough discussion, we entirely agree with your viewpoint, especially the statement "Of course, the obtained TR has larger ranks than the ring in the original TW". From such a perspective, these tensor topologies having an internal core tensor, e.g., Tucker [R5] (ring rank being 1), and projected entangled state pairs (PEPS) [R6], may be able to establish a relationship to TR decomposition, which has been barely studied in previous works.
> > > Moreover, we are sorry that we misunderstood your comment earlier and did not emphasize this exciting property in the current version. Inspired by your affirmation, we will exert more effort and look forward to developing a more comprehensive work in the future, thereby making significant contributions to the tensor network community. Also, we will revise corresponding description to this part, making it clearer to readers.
> > > Again, we gained a lot from your discussions, and sincerely thank you for your remarkable efforts in improving our manuscript.
> > >
> > > [R1] Tensor ring decomposition with rank minimization on latent space: An efficient approach for tensor completion. AAAI, 2019.
> > >
> > > [R2] Fully-connected tensor network decomposition and its application to higher-order tensor completion. AAAI, 2021.
> > >
> > > [R3] Efficient low rank tensor ring completion. ICCV, 2017.
> > >
> > > [R4] Adaptive tensor learning with tensor networks. In Proc. NeurIPS 1st Workshop on Quantum Tensor Networks in Machine Learning, 2020.
> > >
> > > [R5] Some mathematical notes on three-mode factor analysis. Psychometrika, 1966.
> > >
> > > [R6] Matrix product states, projected entangled pair states, and variational renormalization group methods for quantum spin systems. Adv. Phys., 2008.

---

### Official Review · Reviewer_Bfwm · 2022-07-10

**Rating:** 4
**Confidence:** 4
**Soundness:** 2 fair
**Presentation:** 3 good
**Contribution:** 2 fair

**Summary:**

The paper proposed a new tensor network decomposition method, namely Tensor Wheel Decomposition. Different from prior tensor train or tensor ring decomposition, the tensor wheel decomposition introduced a new core tensor factors C in the factorization. An ALS algorithm (algorithm 1) and proximal solution (algorithm 2) are provided. Empirical study in completion task on synthetic data and real data shows that the new algorithm is better than the TT and TR based factorization methods.

**Questions:**

[Q1] In the experimental results of Table1, since the rank are learnt different based on different algorithm, could the author list the rank parameter or the model size to understand the model complexity from different algorithm?

[Q2] The tensor wheel seems to be more general as a combination of tucker decomposition and tensor ring decomposition. Could the author highlight the difference between the TW to Tucker decomposition, and include tucker decomposition as a comparable here?

[Q3] In Figure 3, the data is generated via Tucker decomposition while the comparison is mainly between TT, TR, and TW. Will generating data via TW or TR with Tucker decomposition included, makes more sense for algorithm comparison? Additionally, from the systematic data, it seems TW algorithm is mostly slow and having large performance variance, where TR seems to be faster with smaller performance variance. Will there be any regularization methods to reduce TW performance variance?

**Ethics Review Area:**

["I don’t know"]

**Limitations:**

The paper proposed a new tensor wheel factorization method. Although it is a trivial but interesting extension, comparison to Tucker decomposition method is needed to demonstrate the improvement. From the existing results, model complexity and performance trade-off also needs to provided to demonstrate the efficiency of the new factorization method.  Last but not least, interpretations on the algorithms efficiency and performance variance is needed to show the advantage of TW as compared to TT and TR.

**Strengths And Weaknesses:**

The paper extends the existing tensor networks method to a new tensor factorization. The extension is trivial, but might still brings some new knowledge to the area.

The new tensor wheel factorization seems to be a combination of tensor ring factorization and Tucker factorization, without core tensor taking into considerations. While the comparison to Tucker factorization is not included in the paper.

---

> ### Author Response · Authors · 2022-08-02
> **Rebuttal to all Questions.**
>
> Question 1: Could the author list the rank parameter or model size for each algorithm in Table 1?
>
> Response: We appreciate your comment. Regrettably, we are unable to list all the rank parameters of different algorithms in Table 1, because some of them are non-decompositional, e.g., the HaLRTC [R5] (Tucker decomposition-based convex relaxation method), and TMac-TT [R6] (TT decomposition-based parallel matrix decomposition method). Instead, we have supplemented Tucker decomposition as a comparison method to the synthetic data experiments (see the revised Fig. 3), and wholly listed all their rank parameters (see Table 1 of the revised supplementary material), aiming to compare the model complexity of different algorithms. Consequently, your concerns can reasonably be addressed.
>
> Question 2: Could the author compare TW and Tucker decompositions?
>
> Response: Thanks for your valuable comment. Following your suggestion, we have highlighted the difference between TW and Tucker decompositions (line 166, page 5 of the revised manuscript), as follows: "Unlike Tucker decomposition, TW one considers the potential relationship between adjacent factors and establishes a connection for a higher characterization capacity. Actually, such a strategy also reduces the loadings of core factor, which contributes a smaller ${L_i}, i=1,2,\cdots,N,$ in TW-ranks than Tucker decomposition, thus alleviating the limitations of high storage and computational complexity.". Moreover, since the HaLRTC [R5] is a Tucker decomposition-based method in real-world data experiments, we further included Tucker decomposition as a comparison to the synthetic data experiments (see Tucker-TC (PAM) in the revised Fig. 3).
>
> Question 3: Will generating data via TW or TR with Tucker decomposition included, makes more sense for algorithm comparison? Will there be any regularization methods to reduce TW performance variance?
>
> Response: Thanks for your careful review! In the previous version, we construct the synthetic tensors by Tucker decomposition using Tucker factors, mainly considering that the Tucker factors comprise an underlying high-order structure, which may be closer to reality than TR representation. In the revised Fig. 3, we added the Tucker decomposition as a new competitor for a more comprehensive comparison.
>
> Compared to TR, especially TT decompositions, the proposed TW decomposition has a more complex structure, thus resulting in reduced model stability. When executed on multiple synthetic data using the same parameter configuration, TW-TC inevitably exhibits greater performance variance. Definitely, there are some regularization methods to reduce TW performance variance. As in [R1], by imposing low-rank constraints on the TR factors, the stability of TR decomposition for rank selection is enhanced. Since the high performance variance of TW decomposition is attributable to the TW-ranks, we believe that such a strategy can also be imposed on the TW factors, thereby reducing the performance variance. Notably, the main purpose of this manuscript is to propose a novel tensor decomposition, which shows a more pronounced significance. Hence, we do not consider applying these regularization methods to TW decomposition. They may be more suitable for further work in the future.
>
> [R1] Tensor ring decomposition with rank minimization on latent space: An efficient approach for tensor completion, AAAI, 2019.
>
> [R5] Tensor completion for estimating missing values in visual data, IEEE TPAMI, 2013.
>
> [R6] Efficient tensor completion for color image and video recovery: Low-rank tensor train, IEEE TIP, 2016.

---

> ### Author Response · Authors · 2022-08-09
> **Rebuttal to all Questions -- Round 2**
>
> Thanks for the reviewer's insightful and valuable comments to our work. We have posted all responses in the OpenView system, including the response to your comments. If you have any other questions, you may find the answers from the response to other reviewers. Thank you very much.

---

> ### Author Response · Authors · 2022-08-10
> **Rebuttal to all Questions -- Round 3**
>
> We have detailedly responded to your comments and carefully addressed your concerns. Thanks again for making our results even stronger. Sincerely, we look forward to further communication with you!

---

### Official Review · Reviewer_Fhbz · 2022-07-11

**Rating:** 6
**Confidence:** 4
**Soundness:** 3 good
**Presentation:** 3 good
**Contribution:** 2 fair

**Summary:**

This paper proposes a new tensor decomposition method to improve the downsides of existing ones. The proposed tensor wheel decomposition can be viewed as a hybrid between TR decomposition and Tucker decomposition, i.e., it is similar to TR decomposition but there is an additional core tensor of which the order is the same as the input data and all the factor tensors are connected (tensor-producted) to it. This has some advantages: (1) The Nth order structure of the input data can be preserved. (2) Nevertheless, it is more manageable than FCTN. (3) All the factors are now interconnected by the core tensor, unlike TR decomposition where some factors are far away so it is not easy to find direct relationships between them. Experiments show that, for a few synthetic and real data, the proposed method achieves the best performance.

**Questions:**

The TC experiments only use i.i.d missing conditions which are not realistic. How will it go if we use more realistic conditions, e.g., structured missing conditions?


**Limitations:**

I believe that this paper does not have any serious potential negative societal impact.

**Strengths And Weaknesses:**

The paper proposes a clever way to bypass the downsides of the existing tensor network type decompositions. Unlike TT and TR decompositions, nonadjacent factors are connected via the core tensor and the original structure of the data is better preserved. At the same time, the number of connections is minimized (unlike FCTN) so that it is more manageable. The idea is quite intuitive and convincing, and accordingly, the superior performance is also convincing.

The proposed decomposition is found by a PAM-style method, and the convergence proof is provided. One shortcoming of the paper is that there is no deeper analysis of the decomposition. There are some theorems in the paper, however, they are about basic algebraic properties rather than deeper ones such as approximation error bounds.

Another thing that bothers me is the comparison with FCTN. In Figure 3, Why does FCTN-TC take a shorter time than TW-TC? This seems somewhat unintuitive, considering the main claim (TW decomposition being more manageable than FCTN). Is it because the best performing hyperparameter setting of FCTN happens to be very small? Moreover, it seems that FCTN-TC usually performs worse than TT-TC and TR-TC even though it has heavier connections. Why is this?

---

> ### Author Response · Authors · 2022-08-02
> **Rebuttal to all Weaknesses and Questions.**
>
>
> Thanks for your time and compliments!
>
> Weakness 1: Only provides the basic algebraic properties rather than deeper ones.
>
> Response: Thanks for the insightful comment. Certainly, the deeper analysis of a decomposition, e.g., approximation error bounds, can further contribute to the theories of the manuscript. Nevertheless, such an analysis is usually established on the SVD-based version of the decomposition algorithm, e.g., [R7], rather than the ALS-based version, e.g., [R2, R8]. Since our manuscript only proposes an ALS-based decomposition algorithm, we did not consider providing the suggested analysis before. Following your feedback, we will make efforts to perfect more theories in subsequent supplementary materials, aiming to make further contributions to the tensor community.
>
> Weakness 2: Why does the FCTN-TC in Fig. 3 require less time than TW-TC? Its best hyperparameter setting is small?
>
> Response: You are right! Experimentally, these relatively small parameter configurations almost always yield the optimal performances for the FCTN-TC (see Table 1 of the revised supplementary material), even if the obtained results are unsatisfactory (see the revised Fig. 3), leading to a shorter computing time even than both the TT-TC and TR-TC models. This situation is empirically analyzed in [R2] (lines 31-34, page 3), i.e., the required FCTN rank values are usually far less than Tucker rank values. Moreover, as compared in Fig. 2 of our supplementary material, the proposed TW topology has a more complicated structure than FCTN one under two special cases, i.e., when the operated tensors are third-order or fourth-order. Resultantly, the proposed TW-TC model requires incremental computational time, despite its higher recovery accuracy.
>
> Weakness 3: Why does FCTN-TC perform worse than TT-TC and TR-TC despite larger connections?
>
> Response: According to the optimal rank parameter configuration of the FCTN model, we argue that the inferior performance of the FCTN-TC method may be caused by over-fitting, judged by its several optimal ranks between non-adjacent factors being 2 (see Table 1 of the revised supplementary material). Compared with real-world data, the low-rank characteristics of synthetic data are simpler. That is, not all non-adjacent dimensions have a direct relationship, thus leading the authentic ranks between some non-adjacent factors to be 1 (i.e., without connection). However, the minimum values of FCTN rank among several non-adjacent factors are 2 rather than 1, forcing FCTN topology to maintain its fully-connected structure. Accordingly, some ineffective structures may reduce the performance of the FCTN-TC method, when applied to the synthesized fourth-order and fifth-order data.
>
> Question 1: TC experiments use unrealistic i.i.d missing conditions. How will it go if we utilize structured missing conditions?
>
> Response: This is a good question! To the best of our knowledge, the i.i.d. missing condition is more classical and widely used in tensor completion, e.g., [R1, R2, R4, R8, R9], leading to a more comparative and convincing evaluation, without loss of generality. Actually, the TC experiments using the structured missing condition have previously been verified on RGB images. Although the proposed TW-TC is invariably superior to other compared methods, the more challenging condition leads to inferior performance of all methods. To improve the numerical results, some preprocessing operators, e.g., VDT [R10], or regularizers may be required for these tensor network decomposition-based methods.
>
> [R1] Tensor ring decomposition with rank minimization on latent space: An efficient approach for tensor completion, AAAI, 2019.
>
> [R2] Fully-connected tensor network decomposition and its application to higher-order tensor completion, AAAI, 2021.
>
> [R4] Nonconvex Low-Rank Tensor Completion from Noisy Data, NeurIPS, 2019.
>
> [R7] Tensor-train decomposition, SIAM JSC, 2011.
>
> [R8] Tensor ring decomposition, ArXiv Preprint, 2016.
>
> [R9] Adaptive tensor learning with tensor networks. In Proc. NeurIPS 1st Workshop on Quantum Tensor Networks in Machine Learning, 2020.
>
> [R10] High-dimension tensor completion via gradient-based optimization under tensor-train format. Signal Processing, 2019.

---

> > ### Comment · Reviewer_Fhbz · 2022-08-06
> > **Thank you for the answers.**
> >
> > Thank you for the detailed answers.
> >
> > - I can understand the authors' point about the deeper analysis.
> >
> > - About the above downside of FCTN-TC, I'd like to suggest including the discussion in the paper.
> >
> > - In the rebuttal, the authors said that the performance is decreased for all methods under structured missing conditions. Does TW-TC still achieve the best performance? Could the authors provide some results?
> >
> > --> I'm not trying to say that this point is crucial for the paper's contribution (i.e., this setting is not only about the choice of factorization but also about that of the regularizer, as the authors also stated), but the authors' answer suggests that there are some results, so I'm curious.

---

> > > ### Author Response · Authors · 2022-08-07
> > > **Rebuttal to the further Comments.**
> > >
> > > Thanks for your feedback and the positive comments on our rebuttal. We further respond to your two comments, as follows:
> > >
> > > Comment 1 (i.e., Weaknesses 2-3): Regarding the answered downsides of FCTN-TC, I would suggest including a discussion in the paper.
> > >
> > > Response: Thanks for your valuable comment! Certainly, these two involved discussions relating to Weaknesses 3–4 deserve to be added to this manuscript, aiming at contributing more insightful analysis. Following your suggestion and considering the page limitations, we have incorporated them into the revised supplementary materials (see Appendix C.2). Again, thanks for your efforts in improving our manuscript.
> > >
> > > Comment 2 (i.e., Question 1): Could the authors provide some results?
> > >
> > > Response: For this mentioned experiment, we employ a RGB image with structural missing entries, i.e., the random stripes missing for the color Barbara image of MATLAB, to evaluate the performance of all compared methods. Under this case, the so-called sampling rate amounts to 0.8, i.e., SR = 0.8. Numerically, the results are presented as follows:
> > >
> > > Method: Observed, HaLRTC, t-SVD, TMacTT, TRLRF, FCTN-TC, TW-TC;
> > >
> > > MPSNR:13.47, 13.47, 13.47, 13.47, 21.55, 21.42, 21.91;
> > >
> > > Consequently, we claim that the proposed TW-TC model is invariably superior to other compared methods, and all the performances are diminished. Particularly, since the random stripes missing promote the low-rank property of image data, the three low-rank-based methods, i.e., HaLRTC, t-SVD, and TMacTT, are ineffective. To improve the numerical results, some preprocessing operators, e.g., VDT [R10], or regularizers, e.g., total variation (TV), may be required for these tensor network decomposition-based methods. When we attempted to impose the VDT preprocessing or TV regularizer into the TW-TC model, its performance was improved by 6-8 dB.
> > >
> > > [R10] High-dimension tensor completion via gradient-based optimization under tensor-train format. Signal Processing, 2019.

---

> > > > ### Comment · Reviewer_Fhbz · 2022-08-08
> > > > **Thank you for the detailed answers.**
> > > >
> > > > Thank you for the detailed answers. I'm satisfied with the authors' response.

---

### Official Review · Reviewer_CwGm · 2022-07-11

**Rating:** 6
**Confidence:** 4
**Soundness:** 3 good
**Presentation:** 3 good
**Contribution:** 3 good

**Summary:**

This paper presents a new tensor network decomposition model for high-order tensor analysis. The proposed model factorizes a high-order tensor into a set of latent factors connected to a specific wheel topology. This can be viewed as a combination of the classic Tucker decomposition and the tensor ring decomposition. The numerical experiments of tensor completion application on both synthetic data and real-world examples demonstrate the superiority of the proposed method in terms of reconstruction accuracy.

**Questions:**

- Considering the numerical results in Table 1, why are the reported PSNR values for competing methods TRLRF [A] and  FCTN-TC [B] different from those reported in Table 1 of [B] on the same data (container, news, and HSV)?
- Why does the most complicated model FCTN-TC run much faster than others (e.g., 26.27s v.s. 154.67s) in Toy data

[A] Tensor ring decomposition with rank minimization on latent space: An efficient approach for tensor completion, AAAI 2019
[B] Fully-connected tensor network decomposition and its application to higher-order tensor completion, AAAI 2021

IMO, this paper serves as a nice contribution to the tensor decomposition domain and should be accepted if the above question could be properly answered.
***
**post-rebuttal: I keep my original score though I feel the hyperparameter sensitivity issue should be extensively studied --- I'm not sure whether the superior results beyond other competing methods are partially owing to the test data selection and hyperparameter tuning. Missing to reveal this problem would definitely make the results weaker**

**Limitations:**

N.A

**Strengths And Weaknesses:**

[Strengths]
- Innovative construction of the tensor network model inheriting the merits of the classic Tucker decomposition model
- State-of-the-art performance on real-world tensor completion applications
- Linear scaling for the number of hyperparameters with increased tensor dimension

[Weeknesses]
- Increased computational burden compared with tensor-train and tensor-ring models
- Expensive storage burden scaled exponentially with the dimension of tensor data -- sacrificing one of the significant benefits of the tensor-train model

---

> ### Author Response · Authors · 2022-08-02
> **We carefully answered two questions from Reviewer CwGm.**
>
> We sincerely appreciate your efforts and objective reviews!
>
> Question 1: Why are Table 1's TRLRF [R1] and FCTN-TC [R2] results different from those of FCTN-TC [R2] on the identical data (container, news, and HSV)?
>
> Response: Specifically, the cause consists mostly of two factors: 1) different data, and 2) distinct hyper-parameters. Firstly, the "container" and "news" CVs in [R2] are of size $144\times176\times3\times50$ (50 frames), which is a cropped version of the full size $144\times176\times3\times300$ (300 frames). Although the proposed TW-TC method can also achieve excellent results when tested on an identical 50-frame data, manually tuning parameters for all compared methods is time-consuming, which may cause missing the submission deadline for manuscript. Therefore, we only employ the first 20 frames of CVs for a reduced workload, thus forming our "container" and "news" data sized $144\times176\times3\times20$. Similarly, the HSV in our manuscript is also distinguished from the one in [R2]. Secondly, we have fine-tuned the hyper-parameters of all the compared methods, including TRLRF [R1] and FCTN-TC [R2], resulting in distinct parameter configurations. Consequently, these two aspects collectively contribute to the numerical difference.
>
> Question 2: Why does the FCTN-TC model run quicker in Toy data?
>
> Response: When applied to third-order tensors, the complicated FCTN decomposition degenerates graphically into TR decomposition, i.e. a complete graph with three nodes. Since the "Toy" data is a third-order tensor, the FCTN-TC model for "Toy" data is essentially the TR-TC model, thus allowing for efficient computation. Compared with the FCTN-TC (i.e., TR-TC) model, the TRLRF [R1] model applies extra regularizers for factor matrix rank minimization, while our TW-TC model increases a core factor structure. Moreover, the TMacTT [R3] algorithm experimentally exhibits slower convergence, resulting in a greater number of iterations. Consequently, the FCTN-TC method runs much faster than others, i.e., TMacTT, TRLRF, and TW-TC, in the third-order tensor data, i.e., "Toy".
>
> [R1] Tensor ring decomposition with rank minimization on latent space: An efficient approach for tensor completion, AAAI, 2019.
>
> [R2] Fully-connected tensor network decomposition and its application to higher-order tensor completion, AAAI, 2021.
>
> [R3] Efficient tensor completion for color image and video recovery: Low-rank tensor train, IEEE TIP, 2017.

---

> > ### Comment · Reviewer_CwGm · 2022-08-08
> > **Thanks for answer**
> >
> > After reading the authors' response to Q1, I still have concerns about it.
> >
> > It looks like the proposed method and other competing methods are quite sensitive to the hyperparameter selection, which suggests some degree of overfitting when tuning hyperparameters on certain data.
> >
> > Could the authors provide the results on 50-frame data using the hyperparameters tuned for 20 frames? Does it still lead to better results than other methods reported in [R2]?

---

> > > ### Author Response · Authors · 2022-08-08
> > > **Rebuttal to the further Comments.**
> > >
> > > We appreciate your feedback regarding our rebuttal, and further respond to your one comment, as follows:
> > >
> > > Comment 1: The proposed method and competing methods are sensitive to hyperparameter selection, which suggests overfitting when tuning hyperparameters on certain data. Could authors provide 50-frame results using 20-frame hyperparameters? Does the performance surpass that of [R2]?
> > >
> > > Response: You are right! Except for the HaLRTC method, the remaining methods are relatively sensitive to the selection of hyperparameters. That is, their optimal parameter configurations vary with different data. Therefore, we fine-tune their parameters to not be static, for better performance and fairer comparison. Following your comment, we provided the results on a 50-frame data (i.e., "news" data with SR = 0.1) using those hyper-parameters tuned for 20-frame data (i.e., "news" data with SR = 0.1), as follows:
> > >
> > > Method: Observed, HaLRTC, t-SVD, TMacTT, TRLRF, FCTN-TC, TW-TC;
> > >
> > > Case A: 9.04, 19.58, 28.75, 29.21, 29.31, 30.27, 32.55;
> > >
> > > Case B: 9.05, 20.24, 30.89, 29.48, 32.99, 30.75, 32.89;
> > >
> > > Case C: 8.95, 18.51, 28.13, 27.49, 27.82, 29.52, none (because TW-TC is not involved in [R2]);
> > >
> > > The above Case A, Case B, and Case C, represent the results reported in our manuscript (i.e., 20-frame "news" data with SR = 0.1 using fine-tuned parameters), the requested experiment (i.e., 50-frame "news" data with SR = 0.1 using previously specified parameters), and [R2] (i.e., 50-frame "news" data with SR = 0.1), respectively. Compared with Case A, the performance of all compared methods in Case B are improved owing to the increased data redundancy from 50-frame data. Notably, the TRLRF method [R1] achieved the best incremental improvement (an even higher MPSNR value than the proposed TW-TC model) due to its stability for rank selection. Consequently, when executed on 50-frame "news" data (i.e., Case B), this experiment still yields superior results than other methods reported in [R2] (i.e., Case C). Regarding the difference between Case B and Case C, we consider that it may be caused by two aspects: 1) the different positions of the 50-frame data inside the original data; 2) the fineness of parameter adjustment. However, we emphasize that the parameters of all compared algorithms are finely tuned in our manuscript, thus encountering laborious hyper-parameter selection work.
> > >
> > > [R1] Tensor ring decomposition with rank minimization on latent space: An efficient approach for tensor completion, AAAI, 2019.
> > >
> > > [R2] Fully-connected tensor network decomposition and its application to higher-order tensor completion, AAAI, 2021.

---

### Author Response · Authors · 2022-08-08
**About rebuttal.**

We have responded in detail to each reviewer's comments, but we have not received feedback on our rebuttal from the other two reviewers yet. There are only two days left until the rebuttal deadline, and we hope that our rebuttal will receive an objective and fair response.

---

### Meta-Review · Area_Chair_xwGc · 2022-08-28

**Recommendation:** Accept
**Confidence:** Certain

**Metareview:**

Two reviewers consider that the proposed construction is clearly innovative.
and all reviewers consider that the contribution is useful to the tensor learning community.

The experiments show that the proposed method yields improved performance.

The three reviewers who participated in the discussion with the authors and/or took into account the rebuttal of the author expressed that they were satisfied with the rebuttal.

Reviewer Bfwm who is the only reviewer assigning a score of 4 or lower, did not consider the rebuttal and did not respond to any message after the initial review.
The AC considers that his concerns have been well addressed by the authors and this reviewer states in their initial review that this work "might still brings new knowledge to the area".

The authors are encouraged to take into account in particular the fruitful discussion with reviewer Dt2K to enhance their manuscript with additional discussions and insights, and to further strengthen their experiments if possible (consider core tensors in Tucker with different ranks in different modes), given that the results tend to be sensitive to the choice of hyperparameters (and possibly on the hyperparameter search strategy) and to the choice of dataset.

**Award:**

No

---

### Decision · Program_Chairs · 2022-09-14

Accept